# Alexa Arena: A User-Centric Interactive Platform for Embodied AI

**Qiaozi Gao**[*]    **Govind Thattai**[*†]    **Suhaila Shakiah**[*]    **Xiaofeng Gao**[*]    **Shreyas Pansare**

**Vasu Sharma**[†]    **Gaurav S. Sukhatme**    **Hangjie Shi**    **Bofei Yang**[†]    **Desheng Zhang**

**Lucy Hu**    **Karthika Arumugam**    **Shui Hu**[†]    **Matthew Wen**    **Dinakar Guthy**

**Cadence Chung**    **Rohan Khanna**    **Osman Ipek**    **Leslie Ball**    **Kate Bland**[†]

**Heather Rocker**[†]    **Michael Johnston**    **Reza Ghanadan**[†]

**Dilek Hakkani-Tur**[†]    **Prem Natarajan**[†]

Amazon Alexa AI

## Abstract

We introduce Alexa Arena, a user-centric simulation platform to facilitate research in building assistive conversational embodied agents. Alexa Arena features multi-room layouts and an abundance of interactable objects. With user-friendly graphics and control mechanisms, the platform supports the development of gamified robotic tasks readily accessible to general human users, allowing high-efficiency data collection and EAI system evaluation. Along with the platform, we introduce a dialog-enabled task completion benchmark with online human evaluations.

## 1   Introduction

A longstanding goal of AI is to develop autonomous robotic agents that can assist humans in day-to-day activities. Experiments with robots are often conducted in controlled environments, limiting their variety and scale of operation. To mitigate this problem, several EAI simulation platforms have been proposed, which support virtual scenes that can be either manually designed, synthetically generated, or captured from real scenes. An embodied agent can freely navigate and interact with objects in these scenes to complete tasks. However, the current EAI platforms suffer from a set of limitations that curtail the ability to build generalizable assistive AI agents.

**Facilitating HRI data collection by gamification.** A persistent challenge in Human-Robot Interaction (HRI) is collecting human interaction data comprising of natural language instructions along with visual content and actions. Currently available EAI platforms, however, are not designed for humans to effectively interact with the agent, making the data collection process expensive and time consuming [21, 35]. Games have been historically utilized to encourage wide-spread user participation and engagement with services [6, 49, 30]. To ease the data collection effort, we introduce gamification to EAI, which is achieved by presenting tasks or user interactions in the form of games,

---

[*]Equal contribution. Correspondence to `arena-admins@amazon.com`
[†]Work done while at Amazon Alexa AI

37th Conference on Neural Information Processing Systems (NeurIPS 2023) Track on Datasets and Benchmarks.

and by introducing scoring mechanisms, achievements and streaks to stimulate and engage users, promoting user participation.

**Reasoning based on visual observations.** Tasks in current EAI platforms have limited in-class variability: each task type requires the agent to make use of the same objects repeatedly. As a result, agents may resort to over-engineering (over-fitting to specific and simplistic tasks) and memorization to converge to trivial and non-generalizable solutions, or use predefined task decomposition mappings [22]. One way to mitigate this is to design missions that can be completed in multiple ways by interacting with varied objects in the environment with compositional and causally interconnected state changes, forcing the agent to then use common-sense reasoning to understand its current state from visual observations, past actions and associated state changes.

**Reconciling offline and online execution scenarios.** Real-world operation of assistive agents is characterized by object interactions and collaborative engagement with other agents or humans in the form of visual cues or dialog. In the offline setting, the agent passively follows human instructions. Most currently proposed dialog guided interactive benchmarks and platforms operate in this fashion [11, 2, 26]. Online execution features an agent interacting actively with users that have control, interfacing through natural language in a continuously evolving environment. For example, if the agent veers off its path in the online phase, the human user can regain control of the situation and guide the robot to success, sometimes accomplishing some of the the tasks themselves, contributing a learning opportunity for the agent [37]. Such a corrective mechanism that incorporates the elements of real-time user interaction can be implemented limitedly in an offline setting, to the extent of augmenting datasets with interactive elements like dialogs. With our platform, we aim to bridge the fundamental operational gap between the offline and online scenarios by demonstrating the use of offline interaction data to bootstrap an EAI agent, following which runtime data and interactions could be used to quantitatively and qualitatively improve user experience and task execution success rates, leading to a sucessful demonstration of a human-in-the loop EAI system that more closely resembles real-world deployment scenarios.

To this end, we propose Alexa Arena, a user-centric EAI platform. For better user experience, the platform design has commonalities with games, with features like task-guiding UI elements and engaging visual effects. The platform boasts many variations of objects and state transitions, on top of which a variety of missions are designed (Figure 1). We show an example use case of Arena on dialog-guided task completion, where the embodied agent can communicate with the user through natural language to finish an indoor mission. To assist in developing models for this task, we release a dataset in which each mission is annotated with an expert demonstration and corresponding human-annotated language instructions and dialogues. To evaluate the dialog-guided agent in real-time, we also release a web-based user interface (UI), where a user can communicate with the agent in text and receive visual observations of the agent from Arena. This paper makes the following contributions:

1. Alexa Arena as a new user-centric EAI platform which focuses on building assistive conversational agents that can assist humans through reasoning and procedural learning.

2. A dataset of over 9K dialog sessions and 46K human annotated instructions. Each data session simulates a human user communicating with an embodied agent to complete a household activity.

3. In addition to offline evaluation of the dialogue-guided embodied agents, we also provide an interactive evaluation protocol. Using the web-based interface, the user can provide instructions to the agent and observe the progress in real-time. We also present results from baseline models for both evaluations.

## 2   Related Work

**Embodied AI platforms.** In recent years, many platforms have been proposed for embodied agents to perform household activities in indoor virtual environments [28, 10, 47]. Most simulators are designed with performance and realism as the top priorities [38]. Although some simulators have built-in infrastructure for human data collection [12, 46], the user interactions are generally not the focus of their designs. On the contrary, Alexa Arena is a user-centric embodied AI platform with features designed specifically to enhance user experience. There are also several EAI platforms that are inspired by video games [19, 4, 39, 7, 13, 48], but the agent observations are often simplified.

**Language-guided navigation and task completion.** Most existing work for learning language-guided embodied agents focuses on navigation tasks [3, 36, 24, 9, 31]. For increased task complexity,

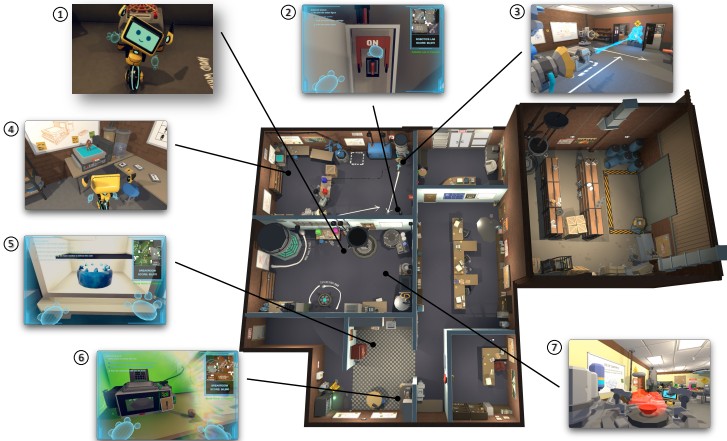

Figure 1: The Alexa Arena EAI platform has a variety of object categories, like a set of fantastical objects as engagement enhancers, such as the freeze ray (3), the time machine (6) and the color changer (7). The agent (1) can make use of different tools to change the state of the object. For example, the agent can use either the freeze ray (3) or the fridge (5) to cool an object. Arena also has more interaction actions and state changes compared to existing platforms. For instance, the agent can use the time machine (6) to repair broken objects, or use the color changer (7) to change the color of objects.

[23, 33] enable the agent to follow natural language instructions and complete household activities, where both navigation and object manipulation actions are required. Most recently, [26, 11] propose new datasets and benchmarks for training and evaluating task-oriented embodied agents that can engage in dialogue. However, both works use offline settings where the dialogues are pre-collected from humans or generated with templates. In comparison, the Arena platform enables embodied agents to communicate with human users in a real-time interactive fashion.

**Task planning with large language models.** Recently, there is a growing trend of using large language models (LLMs) for assisting robot task planning in learning novel activities or completing complex tasks [1, 44]. LLMs have been shown to be good at providing high-level semantic knowledge about the physical world and common human activities [16, 34]. When combined with the sensory perception of the embodied agent, such knowledge can often substantially improve the agent's capability of solving complex tasks in unseen scenarios [18, 17]. The Arena platform provides a good testbed for this line of work. With numerous object types, properties and state changes, along with various environmental causal events, Arena supports the creation of robot tasks that require adequate reasoning capability on common world knowledge.

# 3 The Arena Platform

In this section, we describe the Alexa Arena Platform. We briefly describe the attributes of the simulator and tools that we are releasing as a part of the platform.

## 3.1 Objects Properties and States

There are 336 unique objects in Arena. Each object has a set of properties, called affordances, which specify if a certain type of robot-object interaction is possible. For example, the agent can toggle the *3-D printer* since it has an object property *toggleable*. At the same time, the agent cannot pick it up since it does not have the *pickupable* property. In total, there are 14 object properties, including *pickupable, openable, breakable, receptacle, toggleable, powerable, dirtyable, heatable, eatable, chillable, fillable, cookable, decor* and *infectable*. Each object property has a corresponding action and a subsequent object state upon taking that action. For example, *break* is the corresponding action for *breakable*, and *broken* is the corresponding state after the action has been performed. The states of objects will change as a consequence of the action, given that the pre-conditions are met. For example, if powered on (Fig. 1 (2)), the *3-D printer* can be used to make toys (Fig. 1 (4)).

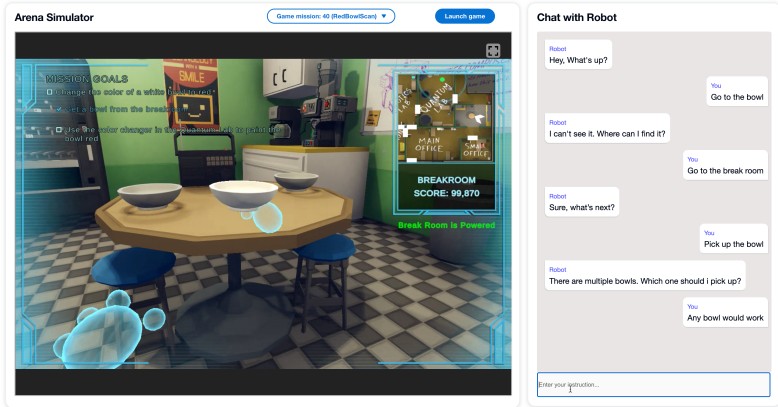

Figure 2: The web-based user interface. The user can use the chat box to communicate with the agent in Arena for task completion. Meanwhile, video from Arena is streamed in real time to show the progress. The minimap is displayed on the top right corner of the game UI, showing the room layout and the robot location and orientation. The mission goal and subgoals are displayed to the users in text on the top left corner of the UI.

## 3.2 Robot Action Space

In general, Arena supports two kinds of actions: 1) user interaction actions for communicating with the user, and 2) robot physical actions to interact with the simulation environment. There are two types of robot physical actions - navigation actions and object interaction actions. For better user experience, all the navigation and interaction actions are animated in a continuous fashion with associated appropriate environmental sounds also being played during the animation.

**User interaction.** To communicate with the user, the robot can initiate a dialog, the contents of which are displayed on the user interface (Figure 2). The robot can also *highlight* objects for real-time visual feedback and use it for confirmation or user instruction disambiguation.

**Navigation.** The goal of Arena is to aid in compositional task learning and reasoning, instead of indoor navigation. To this end, we simplify the navigation in Arena by enabling the robot to directly navigate to a viewpoint in a room by specifying the viewpoint name or the room name, or to an object by specifying the object mask. Meanwhile, if preferred, the robot can also perform step-by-step navigation by a combination of parametrized local primitive actions like *MoveFoward* and *Rotate*.

**Object interaction.** Arena supports 11 actions for object interaction, including *Examine, Pickup, Place, Open, Close, Break, Pour, Toggle, Fill, Scan* and *Clean*. Each action is associated with a set of objects in a specific state in which the objects can afford that action to be performed upon them. E.g., *toggling* can be performed on a *Time Machine* when it is in a *closed* state. Interaction actions are accompanied by a change in the associated object's state, which gets updated in the metadata.

## 3.3 Enhanced User Experience

We incorporate into Arena elements that enhance application usability and improve engagement quality: 1) the platform includes engage-enhancers, such as fantastical objects and a scoring system for each mission; 2) for better visual effects, actions in Arena are animated in a continuous played-out fashion for both manipulation and navigation actions, makes the scenes more natural, user-friendly and engaging; 3) the platform features a unique user interface with several elements to provide users with better task guidance, including minimap and sub-task hints, all of which work in an integrated fashion to improve user-experience.

## 4 Dialog-guided Task Completion

We now present the Dialog-guided Task Completion benchmark. The benchmark is designed to evaluate dialog-guided agents for indoor object interaction tasks. To support model development, we release a hybrid dataset where ground-truth robot action trajectories are paired with human annotated language. We also set up two evaluation protocal: an offline evaluation on the validation split of the dataset and an online evaluation with human users.

| Tasks | Heat & Deliver | Freeze & Deliver | Repair & Deliver | Color & Deliver |
|---|---|---|---|---|
| Scene # | 101 | 105 | 110 | 104 |
| Objects | Bowl, microwave, desk | Mug, freeze ray, table | Floppy disk, time machine, fridge | Apple, color changer, desk |
| Instructions | Take the bowl | Go to the freeze ray target | Carry floppy disk to time machine | Go to the color changer station |
| Questions | Where is the bowl? | What does the target look like? | Where is the time machine? | Where is the station? |
| Answers | On the tan table next to the fan | It is the blue tray attached to the wall | The time machine is in the corner of the countertop | To your left in the southwest corner of the room |
| Expert demonstrations |  |  |  |  |

Figure 3: Examples in the `AID` dataset. Each data session corresponds to one language annotation on a mission that needs to be completed in a specific scene. Each language annotation includes a sequence of instructions guiding the robot to complete the task, as well as questions and answers for clarification. Expert demonstrations are provided for each data session.

## 4.1 Task settings

Each task in the benchmark is specified by the initial states and the goal states of the scene, and a sequence of human instructions. The agent is required to understand the language instructions and interact with the environment through a series of object manipulation and navigation actions to achieve the goal states.

## 4.2 Arena Interactive Dialogue (AID) Dataset

To enable training and evaluation of the dialog-guided embodied agents, we collect the Arena Interactive Dialogue (AID) dataset via crowd-sourcing. The dataset contains expert action trajectories, human language instructions, and associated questions and answers for missions to simulate dialog. We split the dataset into training and validation folds. There are 2661 tasks in training and 383 in validation, in which each task is annotated by 3 annotators. Each human annotation is considered one session, giving total of 7983 training sessions and 1149 validation sessions. There are in total 12 unique tasks types in the dataset. Examples data points are shown in Figure 3.

**Expert demonstrations.** The game mission and expert demonstration generation follows a two-step process. In step one, game missions are programmatically generated via sampling from initial environment states and mission goals. The second step generates expert demonstrations using a planner. Along with the game definitions, we also generate PDDL (Planning Domain Definition Language) planning problem definitions for each game mission. A symbolic planner is used to solve the planning problem and the output action sequence is collected as the expert demonstration. One thing to note is that the planner has access to the game metadata, which is not available to the agents during inference time. For tasks that can be completed in different ways (e.g., by using different tools), we pick one unique way for each task to generate expert demonstration. For example, for freeze and deliver, in some missions the fridge is used to cool the object, while in other cases the freeze ray is used.

**Human Language Annotation.** To collect natural language dialogue for the missions, we design a two-stage data annotation process on Amazon Mechanical Turk (AMT). In both stages, the annotator watches a video containing an expert demonstration for the mission and provides the annotations in free-form text data or answers to multiple choice questions. In the first stage, annotators write instructions to tell a "smart robot" how to accomplish a task. During the process, an annotator first watches the video of the ground-truth robot actions, then writes instructions for each highlighted video segment. After all the instructions are collected, we start the second stage of annotation, where the annotators are asked to raise questions to better complete the task, as if they are controlling the robot to follow the instructions. They also need to subsequently answer their own raised questions. Similar to [11], the question choices in the second stage of annotation are generated using predefined templates. For more details on the human language annotation process, see Appendix.

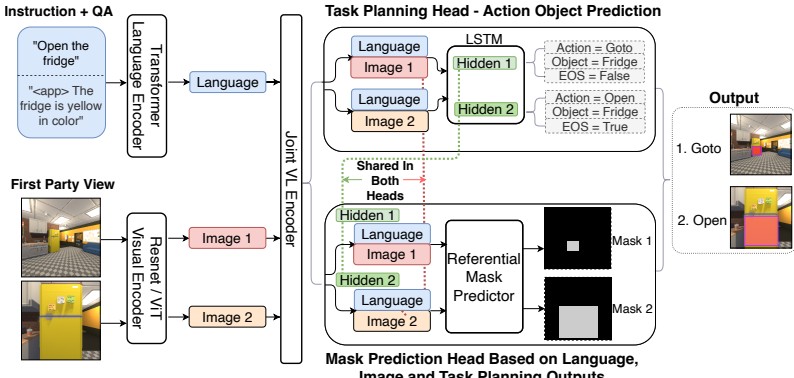

Figure 4: Model architecture of the vision language model for end-to-end action prediction and visual grounding.

## 4.3 Offline evaluation

We expect that a good embodied agent should be able to finish the missions efficiently. To achieve this goal, the agent should understand the human instructions and generate a corresponding sequence of actions. Thus, we evaluate the agent using the following metrics:

- **Mission success rate (MSR).** For each mission, there is a mission success variable $m$ indicating whether the goal conditions have been met for the mission. $m = 1$ if all the goal conditions are met. Mission completion rate is calculated by averaging $m$ across all the missions.
- **Average number of robot actions (NRA).** To measure the efficiency of task completion, we also record the number of actions taken by the agent to complete each mission. Average number of robot actions is calculated by averaging the number of actions across all the missions.

## 4.4 Online Human evaluation

To demonstrate the potential of Alexa Arena for real-time user interactions, we design a human evaluation protocol for dialog-guided task completion. Different from the offline evaluation where the agent only passively follows the user instructions, in this online setting, the agent can also engage in multi-turn dialog with the user to communicate its internal states and clarify the mission goals.

**Missions.** On top of the 12 task types presented in the AID dataset, we design and curate 14 more complex longe-horizon missions, each requiring the agent to complete several sub-goals by following the user instructions, aiming to test compositional learning capabilities of the agents. The missions are designed to be gamified to engage the users and the environment also contains hints to guide the users to complete the tasks. The setting encourages dialog, where the human and the robot share the goal to complete the mission. An example of such a task is provided in the Appendix. To test the agent's capability to generalize across tasks, we keep 5 missions as the unseen test set.

**Metrics.** For online evaluation, we use the MSR as the task performance metric. As a subjective evaluation metric, we collect users' overall satisfaction toward the game on a 5-point likert scale after each interaction.

# 5 Experiments

## 5.1 Offline evaluation

We use the AID dataset to train two baseline models. The inputs to the models are the agent egocentric view and natural language conversations. The outputs are a series of executable action sequences and associated interactable masks (where needed) to complete the task at hand. For both models, we experiment using only the human instructions, as well as the instructions appended with the questions and answers (QA) as input to simulate human-agent dialog during the mission in offline evaluation.

**Neural-Symbolic Approach.** Episodic Transformer (ET) is a neural-symbolic model for visual-language task completion [27]. ET uses a multi-modal transformer to predict actions and object pairs based on the visual and language information in the entire episode, with a separate object detector to generate object masks.

| Mission Type | NS w/o QA | NS w/ QA | VL w/o QA | VL w/ QA |
|---|---|---|---|---|
| breakObject | 0.00±0.00 (10.93±0.22) | 0.00±0.00 (9.90±0.05) | 28.88±3.62 (8.90±0.42) | 37.41±3.66 (9.07±0.56) |
| clean&deliver | 14.18±1.43 (21.66±0.06) | 13.79±0.00 (15.14±0.09) | 11.49±2.48 (16.12±0.86) | 16.85±3.01 (13.63±0.93) |
| color&deliver | 0.00±0.00 (14.44±1.43) | 0.00±0.00 (13.22±0.17) | 0.0±0.0 (16.69±0.45) | 2.77±3.92 (18.05±0.67) |
| fill&deliver | 14.58±0.00 (20.38±0.27) | 20.14±0.98 (17.12±0.08) | 11.80±4.28 (15.43±1.27) | 23.61±5.47 (13.79±1.86) |
| freeze&deliver | 31.94±1.96 (17.17±0.07) | 19.44±1.96 (17.42±0.32) | 1.38±1.96 (21.48±1.65) | 5.55±5.19 (19.71±2.79) |
| heat&deliver | 5.13±0.00 (14.79±0.36) | 5.13±0.00 (18.22±0.45) | 23.07±5.53 (19.24±1.97) | 28.20±2.09 (16.91±0.57 |
| insertInDevice | 14.69±0.00 (11.53±0.15) | 14.88±0.53 (11.45±0.04) | 15.63±2.62 (7.54±0.46) | 22.03±1.22 (1.22±0.43) |
| pickup&deliver | 9.47±0.00 (11.15±0.10) | 11.93±0.29 (12.66±0.08) | 21.28±0.92 (8.03±0.36) | 26.54±1.67 (7.89±0.40) |
| pourContainer | 15.10±0.40 (12.19±0.04) | 14.81±1.07 (12.71±0.12) | 25.92±5.04 (11.09±0.90) | 31.05±1.06 (10.07±0.29) |
| repair&deliver | 11.11±0.00 (17.19±0.28) | 12.96±0.00 (15.56±0.35) | 11.72±0.87 (21.69±1.14) | 18.51±3.02 (18.39±1.23) |
| scanObject | 41.44±0.00 (4.90±0.00) | 41.44±0.00 (4.84±0.00) | 41.44±3.20 (3.72±0.19) | 57.65±4.59 (3.96±0.36) |
| toggleDevice | 57.14±0.00 (5.09±0.00) | 56.19±0.00 (5.08±0.00) | 84.12±3.23 (4.13±0.22) | 87.62±4.85 (4.49±0.68) |
| **average** | 18.42±0.00 (11.89±0.01) | 18.97±0.00 (11.62±0.03) | 27.00±0.39 (9.88±0.29) | 33.88±0.71 (9.20±0.28) |

Table 1: Experimental results on the validation dataset for the Neural-Symbolic (NS) and Vision-Language (VL) models. For each task, we show mission success rate and average number of robot actions (in the parenthesis). For each metric, we report the mean and standard deviation from three runs.

**End-to-end Vision Language Model.** We experiment with an end-to-end vision language (VL) model shown in Figure 4. A transformer model encodes the natural language instruction, and the egocentric view is encoded using a ResNet [15], both of which are initialized with pre-trained CLIP weights [29]. The encodings are passed through a joint VL encoder to get a combined vision-language embedding. This is then channelled through two heads: 1) The task planning head is an LSTM module encoding the context of previous predicted outputs, followed by two linear classifiers to predict the *(action, object)* pairs. 2) The mask prediction head uses the current first party view, the encoded language and the action-object prediction hidden state from the task planning head to predict a single mask referring to the interactable object using the referential image segmentation architecture described in [45]. The above procedure is repeated for a natural language instruction and a series of first party views until an End-of-Sequence (EOS) flag is emitted by a separate binary classifier.

### 5.1.1  Results and analysis

The baseline models are trained on the training split, and evaluated on the validation split of the `AID` dataset. The overall mission completion results are displayed in Table 1. Both models are evaluated with a cap of 50 maximum allowed steps and 10 failed steps per mission, beyond which if the mission goal is not completed, the agent stops execution leaving the goal incomplete. Below, we analyze the results for both models and provide insights on the model performances.

**Neural-Symbolic Model.** Overall, adding QA leads to a marginal improvement in the performance for the Neural-Symbolic model. The model performs well in MSR for missions with short horizons (e.g. toggleDevice). As an ablation study, we also evaluate the multi-modal transformer on the validation set by providing the ground-truth visual observations in the dataset at each time step. As a result, the model can correctly predict all the actions and objects for 71.3% of the missions without QA, and 76.9% with QA. Since it is common that there are multiple instances of the same class in a room, the model can choose an incorrect interactable mask even if the transformer model correctly predicts the object class and the vision model correctly predicts all the object masks of that class in the image. This can also explain why the improvement with QA is limited, since the additional information does not participate in the grounding once the action and object pairs are predicted.

**Vision Language Model.** The VL model shows a 6.88% absolute improvement in MSR when trained and evaluated with QA compared to training and evaluating without QA. The agent is also able to complete the missions more efficiently with QA, demonstrating that asking the right questions

| Team | Week 1 | Week 2 | Week 3 | Week 4 (Unseen) | Offline |
|---|---|---|---|---|---|
| SlugJARVIS | 3.98±0.18 (52.82) | 4.10±0.19 (67.35) | 4.19±0.18 (49.56) | 3.90±0.20 (55.56) | (3.66) |
| ScottyBot | 3.48±0.21 (46.46) | 3.67±0.21 (40.40) | 3.97±0.15 (49.04) | 3.63±0.21 (57.95) | (8.36) |
| EMMA | 3.31±0.20 (47.66) | 3.90±0.18 (54.44) | 3.88±0.18 (57.61) | 4.28±0.14 (56.12) | (30.55) |
| SEAGULL | 3.57±0.19 (33.78) | 3.77±0.18 (50.00) | 4.31±0.16 (51.72) | 3.63±0.21 (37.86) | (30.98) |
| GauchoAI | 4.05±0.17 (62.36) | 4.31±0.16 (68.38) | 4.48±0.15 (64.35) | 4.33±0.17 (66.67) | (36.47) |

Table 2: For online evaluation results, we show the weekly user satisfaction on a 5-point scale and missions success rate in percentage (in the parenthesis). We report the mean and standard deviation for the user satisfaction. For each team, we also show the MSR achieved in the offline evaluation for comparison.

improves task execution efficiency and accuracy. For tasks like *toggling* which few objects can afford, the instructions tend to already be descriptive enough in natural language (*"Turn on the red computer", "Press the blue button"*), which explains why adding QA shows a mere 3% improvement in MSR. For all other task types which involve numerous objects, the QAs provide crucial additional information that is not naturally provided with the instruction in the first turn, thus leading to significant performance improvement both in MSR and NRA. From analysis, the *color & deliver* and *freeze & deliver* missions are prone to failure (as evidenced by low MSRs) because of relatively longer contexts than other missions and difficulty in disambiguating the receptacle of *delivery*.

## 5.2 Online Human evaluation

Alexa Arena is used as the interactive EAI platform in the Alexa Prize Simbot Challenge, in which human users can perceive the environment through the agent's egocentric view, and communicate with the embodied agent via natural language commands. To understand how additional user interaction data can help model development, the challenge schedule follows a continual learning setting in a four-week duration: in the first three weeks, the teams can freely update their model using interaction data on 9 missions, and in the final week the models are tested on 5 unseen missions for generalizability.

### 5.2.1 Modeling Methodologies

In total, 10 teams participated in the challenge and implemented their own models for Arena. The models output physical actions for the agent and engages in dialog with users to successfully complete missions. We briefly describe the design of some representative models. SlugJARVIS [43] use a hybrid approach for action prediction, with a rule-based parser to process most requests and an end-to-end Pythia [5] model to handle corner cases. They also design a progressive task experience retrieval module which helps make use of the existing successful user interaction data on seen tasks to enhance performance. ScottyBot [40] design a modularized system containing a neural NLU module for action prediction, named entity recognition, and action/object names standardization. The proposed design also incorporates semantic maps for efficient navigation and object localization. SEAGULL [25] use a rule-based system to detect user intent, and a PDDL planner to generate robot actions. They also design a hierarchical vision model to detect environment states, including fine-grained object categories, object states and objects spatial relations. EMMA [41] propose an encoder-decoder architecture to encode the multi-modal input and generate outputs as textual and visual tokens. The model is first pretrained on multiple vision-language tasks including masked language modeling and image captioning, before finetuning on embodied planning tasks. GauchoAI [42] also use a modular framework for their agent, with a hybrid rule-based and neural action prediction module, an agent state tracking module, a preprocessing module to reject non-plausible actions, and a MaskFormer [8] based visual grounding model. We refer readers to the technical reports for additional details [3].

### 5.2.2 Results and analysis

In Table 2, we show human evaluation results for each team on a weekly-basis. For the first three weeks, we can see a clear upward trend for user satisfaction within each week across all the models. For the unseen missions, most teams are able to achieve similar performance as seen ones, showing the potential to generalize to novel tasks. Compared to the offline performance, we observe that the teams are able to achieve a much higher mission success rate with human evaluation despite the tasks being more complicated. This shows the effects of having interactive dialog during task execution, which helps the agent understand the task goal and the user to understand the model's limitations. This is validated qualitatively by playing with the virtually deployed bots, where we observe critical

---

[3]https://www.amazon.science/alexa-prize/proceedings

contributors to a higher user satisfaction and MSR. While there is a positive correlation between MSR and user ratings across different teams, users tend to give a higher rating to bots that are transparent and collaborative, able to give clear feedback, let their limitations be known, and are good at anticipating user actions in a balanced manner, as opposed to bots taking actions without appropriate feedback. For example, from Week 2 to Week 3 for certain models such as EMMA and GauchoAI, the user ratings increased while the overall MSR decreased. The increase in user satisfaction is attributed to a variety of factors, including that these models became more user-centric, were more interactive and feedback friendly to the users. The teams were able to update their models with real-time human feedback signals to align their bots with the users that interact with them.

## 6 Conclusion and future work

We introduce Alexa Arena, a user-centric EAI platform, featuring user-friendly graphics, animations and control mechanisms to facilitate user-centric EAI research. We provide baseline results for a dialogue-guided task completion use-case, and also release the `AID` dataset. While the Arena platform is designed to improve user-experience for EAI systems, the proposed online models cannot be evaluated in a straightforward fashion without a human-in-the-loop to supply interactive dialog and feedback. To overcome this limitation, we aim to incorporate a model-based user-simulator in future work, to which an oracle provides relevant answers during task execution. Another direction to explore is enabling the human and the robot to control different embodied agents and create more scenarios for HRI.

**Ethical Considerations.** Via introducing Arena as a public embodied AI platform built for human-robot communications, we mainly target English language and do not explicitly look at the issues innate to natural language understanding such as comprehending under-represented languages. The missions, tasks and game designs can be controlled by the platform users as needed, and is backed by a framework that has been validated quantitatively and qualitatively, to the best of our abilities, by carefully constructed and robust user studies as presented in the paper. Since Arena is to be open-sourced, it is possible for users to modify the platform for unintended use cases, which is a common risk for all open-sourced embodied AI platforms. But we believe the risk for Arena is low comparing to other platforms since Arena uses a gamified setting and the released recources are not directly applicable for real-world malicious applications. As for real world transferability of the trained agents, robust testing and continual safety monitoring mechanisms should be put in place to minimize risks of unpredictable robot behaviours in unexpected scenarios.

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

## Supplementary Materials Overview:

- Appendix A.1 contains details of the platform.
- Appendix A.2 contains definitions of tasks.
- Appendix A.3 contains details of data collection process for the `AID` dataset.
- Appendix A.4 contains details of the vision dataset.
- Appendix A.5 contains details of the baseline models.
- Appendix A.6 contains error analysis of the VL model.

Our code and data are available at https://github.com/amazon-science/alexa-arena.

# A Appendix

## A.1 Alexa Arena Platform Details

### A.1.1 Architecture

The architecture of the Alexa Arena framework is illustrated in Figure 5. The core components of the platform are the *arena wrapper* and *model wrapper*. The *arena wrapper* houses the *arena orchestrator*, which takes as input the configuration file (described in Section A.1.2), and communicates with the arena engine, which contains the the *arena simulator* and the *streaming server*. The *streaming server* in the arena engine streams the robot's first party camera view to human users. The *model wrapper* contains the *inference model* and the *model executor* that interfaces between the model and the orchestrator. The *model wrapper* can be modified to develop other models and execution strategies. In addition, we also provide a web interface (the *webtool*) for interacting with a robot model in a chatbot like interface, while also watching the game play out in streaming on a web page.

### A.1.2 Challenge Definition Format

To define the mission configurations, we introduce a schema called Challenge Definition Format (CDF). A CDF file is a json-format file that is readable by Arena. It contains the necessary information to configure, initialize and run a game mission in Arena. These include:

**1. Game initial state:** game scene setting, robot initial location, object initial location and initial states (such as cabinet door being open or closed).

**2. Game goal condition:** robot and object states that need to be satisfied. Once the goal condition is met, the game mission is considered completed.

**3. Game-related text data:** textual data that help guide the user to finish the game, including game mission description text, sub-goal descriptions, system prompts during game play, etc.

The motivation for designing and utilizing the CDF is to allow programmatic generation of large numbers of game missions, which is essential for data collection. Since manually defined game missions are not scalable for generating large-scale training data for ML models, with the easily configurable CDF files, we can generate thousands of game missions by permuting scenes, robot location & states, object types, object location & states, and mission goal states. The other advantage is that since the CDFs are human readable, we can specify new game missions flexibly without writing code.

### A.1.3 Environment Metadata

The environment metadata in Arena contains robot egocentric view images, agent state, objects state, scene metadata, goal condition status, and previous action execution result. To facilitate different task and modeling settings, the robot camera view images support different modalities, including RGB image, depth image, instance segmentation image and normals image. The metadata response from the arena simulator is updated after each action execution.

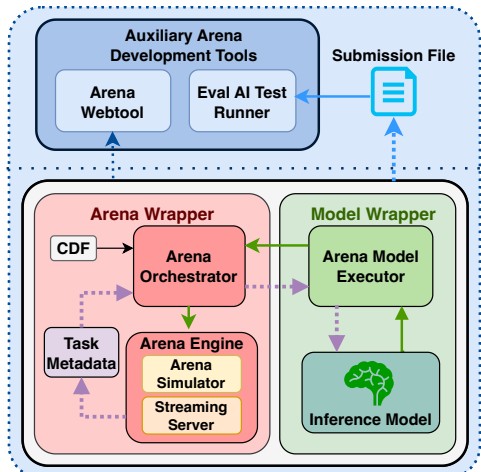

Figure 5: Overview of Alexa Arena Platform Architecture. The core modules are the *arena wrapper* and *model wrapper*. The arena engine contains the *simulator* and the *streaming server*. The *metadata* from the Arena Engine are sent to the inference model, which generates actions to be executed in the simulator. We have also built infrastructure to generate *results submission files* for EvalAI challenges.

### A.1.4 Objects and State Transitions

There are 336 unique objects in Arena. Each object has a set of properties, called affordances, which specify if a certain type of robot-object interaction is possible. To further mimic the complexities of real-world tasks and execution strategies, along with numerous object types and properties, Arena also provides a variety of constraints defined by commonsense causal events. For example, the power system enables a series of causal constraints on using electric appliances, e.g., the fuse box needs to be reset before lights being turned on, the microwave needs to be plugged in to the outlet before it can be powered on. Agents in Arena also have multiple ways to complete a task, e.g., To heat something, the agent can make use of either the *Laser* or the *Microwave*. As a result, the agent needs to reason based on its current state for efficient task completion. As an example, consider the task of heating a bowl and delivering it to a receptacle in the *Quantum Lab*, and the agent is currently in the *Quantum Lab*. If the agent's execution path to heat the bowl using the *Laser* in the *Quantum Lab* is physically obstructed, it is more efficient for the agent to go to the *Breakroom*, heat the bowl quicker using the *Microwave* and return to deliver the bowl in the *Quantum Lab*. A full list of of objects and their properties and state transitions are available in the "arena_documentation.html" file in the supplementary materials.

### A.1.5 Layouts

Alexa Arena contains 10 large game-ready interative multi-room layouts. Each layout features an office-like environment with modular design, allowing the contents within rooms to be rearranged and repositioned. The layouts were hand designed to be visually different and aesthetically interesting. The office setting allows both mundane and fantastical interactive elements: users would be accustomed to seeing everyday objects in more conventional rooms (and would know instinctively how to interact with them), but entering rooms clearly labeled as "labs" they would expect to find futuristic devices and prototype machines that would require experimentation to discover their function. As a result of the above bipolar content, we are able to create game missions with imaginative situations sitting alongside normal everyday objects. The layouts in Arena are large: on average, each house has 190 square meters of navigable area, which is significantly larger than the scenes in other EAI platforms (Figure 6). Figure 7 shows the top down images of 4 houses in Alexa Arena.

### A.1.6 Related platforms

See Table 3 for a comparison between Alexa Arena and other embodied AI platforms.

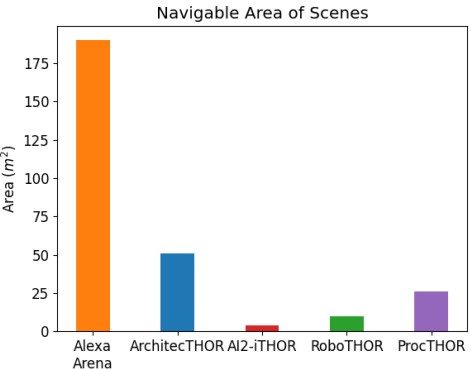

Figure 6: Bar plot of the navigable areas for Alexa Arena, comparing to ArchitecTHOR, AI2-iTHOR, RoboTHOR and ProcTHOR.

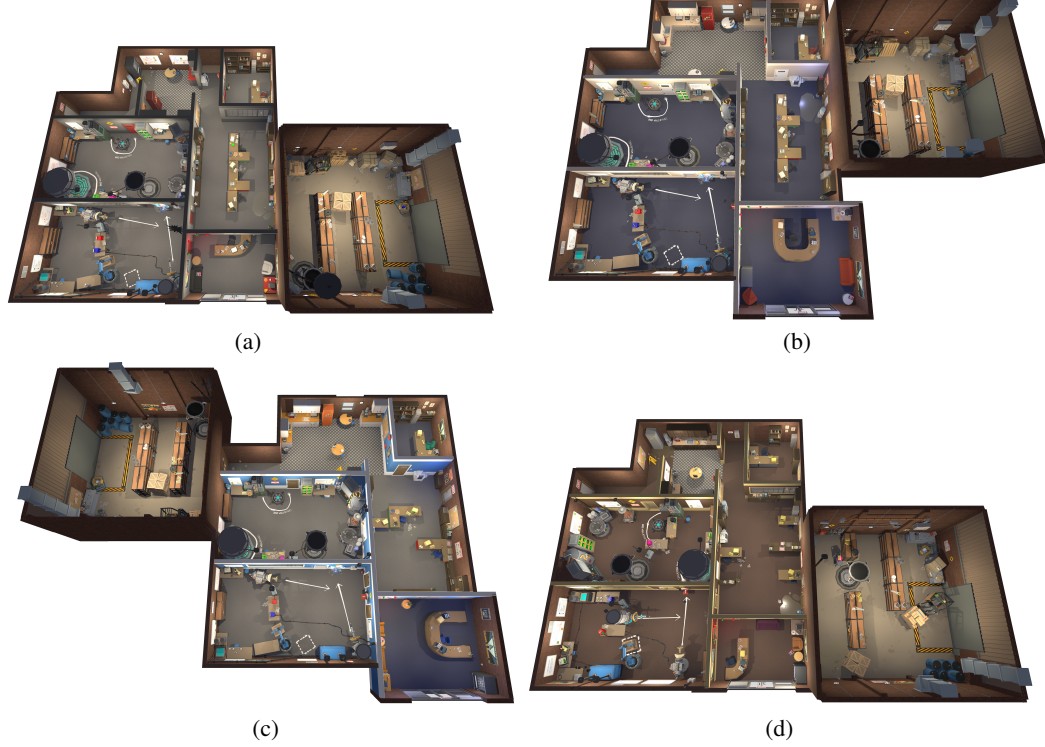

Figure 7: Top-down images of 4 multi-room game-ready layouts in Alexa Arena.

## A.2 Task definition

We designed two set of tasks: 1) the `AID` dataset include 12 simple tasks that require the agent to perform some basic object interactions, 2) the 14 compositional missions are used for online evaluation, each mission requires the agent to complete multiple sub-goals.

### A.2.1 Basic tasks

There are 12 types of basic tasks. For most tasks, the agent is required to find the target object, use a specified tool to change its state and deliver it to the designated receptacle.

**toggleDevice.** The agent is required to find the device and toggle it. For some devices (e.g. color changer), the agent needs to interact with the correct object parts (e.g. red button vs green button) in order to finish the task.

| Platforms | Scenes | Object Types | Multi Room | Object States | 3D Env | Real Scenes | Human Control | Game Ready |
|---|---|---|---|---|---|---|---|---|
| Malmo [19] | 17* | 1000+ | | ✓ | ✓ | | MK | ✓ |
| Overcooked [7] | 6* | 5 | | ✓ | | | MK | ✓ |
| Deepmind Lab [4] | 28* | - | | | ✓ | | MK | ✓ |
| AI2-Thor [20] | 120 | 125+ | | ✓ | ✓ | ✓ | MK,VR | |
| iGibson [46] | 15 | 390+ | ✓ | ✓ | ✓ | ✓ | MK,VR | |
| Habitat 1.0 [32] | 1000 | - | ✓ | | ✓ | ✓ | MK | |
| Habitat 2.0 [38] | 105 | 90+ | ✓ | ✓ | ✓ | ✓ | MK | |
| VirtualHome [28] | 7 | 170+ | ✓ | ✓ | ✓ | ✓ | NL | |
| TDW [10] | 15 | 200+ | | ✓ | ✓ | ✓ | VR | |
| VRKitchen [12] | 16 | 200+ | | ✓ | ✓ | ✓ | VR | |
| **Alexa Arena** | 10 | 335+ | ✓ | ✓ | ✓ | ✓ | MK,NL | ✓ |

Table 3: Comparison with other platforms. Scenes and objects: the number of scenes and interactable object types the platform provide. Multi-room: multi-room scenes. Some platforms marked by * are customizable and new scenes can be added. States: agent can change object states via interactions. 3D env: whether the platform supports 3D virtual environment. Real scenes: whether the scenes in the virtual environment are realistic. Human Control: the interface for the human to control the agent. MK: mouse and keyboard. NL: natural language. VR: virtual reality. Game ready: whether the platform has game-ready features to enhance user experience, including environmental and interaction sounds, action animations, smooth navigation, fantastical objects, hazards, mini-map, scoring mechanism and hints.

**pickup&deliver.** The agent is required to first find the target object, pick it up, navigate to the target receptacle and put the object on the receptacle.

**breakObject.** The agent is required to pick up a hammer, navigate to the target object, and use the hammer to break the object.

**clean&deliver.** The agent is required to pick up the dirty object, navigate to the sink, turn on the faucet and clean the object using the water. The agent can then deliver the cleaned object to the receptacle.

**color&deliver.** The agent is required to pick up the object, go to the color changer and put the object on the color changer. The agent can then push a button on the color changer to change the color of the object, and deliver it to the receptacle.

**fill&deliver.** The agent is required to pick up a container, go to the sink, and turn on the faucet. The agent can then fill the container with water and deliver it to a receptacle.

**freeze&deliver.** The agent is required to pick up the target object and use either the fridge or the freeze ray to freeze the object. The agent can then deliver it to the receptacle.

**heat&deliver.** The agent is required to pick up the target object and use either the laser or the microwave to heat up the object. The agent can then deliver it to the receptacle.

**insertInDevice.** The agent is required to pick up some small object (e.g. floppy disk, printer cartridge, laser component) and insert it into the corresponding device to assemble the device.

**pourContainer.** The agent is required to pick up the container (e.g. milk carton, cereal box, coffee pot), navigate to the receptacle and pour the things in the container into the receptacle.

**repair&deliver.** The agent is required to pick up a broken object, and use the time machine to repair it. The agent can then deliver the intact object to the receptacle.

**scanObject.** The agent is required to go to a place near the target object and scan it.

### A.2.2 Compositional tasks

For the online human evaluation, a suite of 14 long-horizon missions are designed to test compositionality and generalizability. Each mission includes multiple sub-goals that need to achieved to complete the overall task. The human users can use the sticky notes placed in the scene to guide themselves during mission play. An example mission is provided below with the overall setup, goals and subgoals. Only the goals and subgoals are visible to the users during mission play.

```
Scene: The robot is spawned in the Robotics Lab, a trophy is kept
on one of the tables in the corner, but access to that side of the
room is blocked by a wooden block. The block can only be moved by
toggling a robotic arm, which lifts and displaces the block and
clears the path.

☐ Find the trophy and deliver it to the manager's desk.
    ☐ Pick up the trophy
    ☐ Deliver it to the manager's desk
```

If needed, hints are located at visible locations to unblock the user to complete the mission. For the above example, the happy path of the robot to complete the mission without any impediments involves toggling the robotic arm to move the wooden block, gaining access to the other side of the room. The robot then picks up the trophy which is now accessible, and navigates to the manager's desk to deliver it. Such a mission tests the visual and compositional reasoning of the agent. The agent must understand that the path to the trophy is blocked, and that the robotic arm controls the wooden block, which can be moved to clear a path.

The remaining 13 missions are described briefly below:

**Assemble and fire the laser cannon.** The agent is spawned in the quantum lab, is required to find a control panel and place it on the laser (which looks very similar to the freeze ray). The laser is then turned on using the laser monitor (one of two monitors disambiguated by color and position).

**Turn a white bowl red.** The agent is spawned in the quantum lab, is required to pick up one of 3 bowls from the breakroom and use the color changer to change the color of the bowl.

**Disinfect the computer.** The agent is required to find the floppy disk in the manager's office, navigate to the main office and use the disinfecting floppy on the computer with an infection (indicated by a skull on the monitor among other monitors).

**Freeze the soda can with a freeze ray.** The agent needs to find the soda can on a table and freeze it using the freeze ray. This requires finding the right monitor to toggle.

**Defrost the cake with laser.** The agent needs to find a frozen cake in the freezer and defrost it using the laser in the quantum lab, which requires disambiguating between tables and monitors.

**Print action figure.** The agent needs to print an action figure using the 3D printer. Before doing this, the agent needs to turn on the power in the room (as the room is initally dark) to power the 3D printer on.

**Turn a red apple green.** The agent needs to find a red apple in the lower fridge and use the color changer on it to turn it green.

**Make a bowl of cereal.** The agent is required to make a bowl of cereal by pouring milk and cereal into a bowl in the breakroom. The milk is initially in the fridge, the cereal is on the breakroom table, and the bowl is initially broken. The agent needs to fix the broken bowl before making the cereal.

**Pour coffee and deliver to reception.** The agent is required to pour coffee into a mug from the breakroom and deliver it to the reception table.

**Heat a cup of coffee using the laser machine.** The agent needs to find the mug with the coffee and heat it using the laser machine in the quantum lab. This requires finding the laser monitor and toggling it.

**Scan items.** The agent is required to scan a computer, a light switch and a vending machine.

**Find prorotype and toggle.** The agent needs to find the prototype machine with a face on it, and toggle it repeatedly until a sad face appears.

**Print mug.** The agent is required to find the mug printer cartridge on a table, insert it in the 3D printer and use it to print a 3D mug.

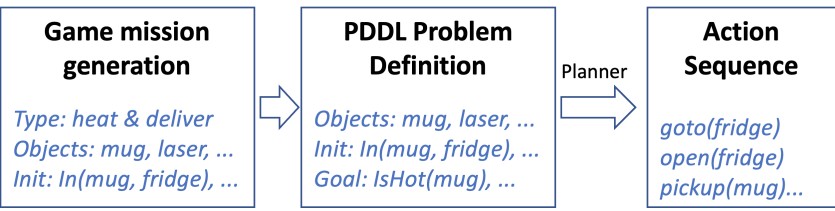

Figure 8: Process of generating game missions and expert demonstrations.

## A.3  `AID` Dataset Collection

In this section, we describe the details of the data collection process for the Arena Interactive Dialogue (`AID`) dataset. The game mission and expert demonstration generation process is illustrated in Figure 8. After generating the missions and expert demonstrations, we use Amazon Mechanical Turk (AMT) for collecting human instructions as well as questions and answers necessary for task completion. Annotators receive $1 compensation for each annotation task which roughly takes 3 minutes to complete, and the hourly wage is around $20. All annotators agree to the Amazon Mechanical Turk Participation Agreement[4] before starting the annotation.

### A.3.1  Human Instruction Annotation

To collect natural language dialogue for the missions, we design a two-stage data annotation process. The first stage is for collecting instructions and the second is for questions and answers. In both stages, the annotator watches a video containing an expert demonstration for the mission and provide the annotations by text data or multiple choice questions.

In the first stage, annotators are told to write instructions to tell a "smart robot" how to accomplish a task. During the process, an annotator first watches the video of the ground-truth robot actions, then writes instructions for each highlighted video segments. Fig. 9a shows the user interface for collecting language instructions.

After all the instructions are collected, we start the second stage of annotation, where the annotators are asked to raise questions to help them complete the task, as if they are controlling the robot to follow the instructions. They also need to answer their own questions. Fig. 9b shows the interface for collecting questions and answers. Here are the steps annotators need to follow:

1. Annotator watches a video: A 10 second video clip is played till the beginning of a sub-task, and the respective instruction is shown on screen. The video clip helps the human annotator to understand the initial environment states.

2. Annotator selects a question: The annotator selects one pertinent question that they think may help task completion. The questions are generated from several predefined templates. Annotators also have the options to type in their own questions.

3. Annotator answers the question: The same annotator watches the video segment of an expert agent working on the sub-task specified by the language instruction, and then providing the answer to the question in text.

4. Annotator indicates whether asking the question is necessary: In some cases, the instruction and the visual context is already clear enough, and asking a question is not necessary.

To ensure the annotation quality, in practice, we collected 3000 data sessions for the first data collection stage. Then we asked additional annotators to identify annotation errors and try to correct them, e.g. erroneous language instructions, grammatical errors, misalignment between instructions with robot actions, etc. For data sessions where most collected annotations do not make sense and are not worth fixing, the additional annotators are asked to simply ignore them. As a result, 20.3% sessions are ignored and 35.7% instructions are corrected. From the data verification results, we identified a subset of AMT workers (around 300 workers) who generated high-quality data, and used them for our subsequent data collection process.

---

[4]https://www.mturk.com/participation-agreement

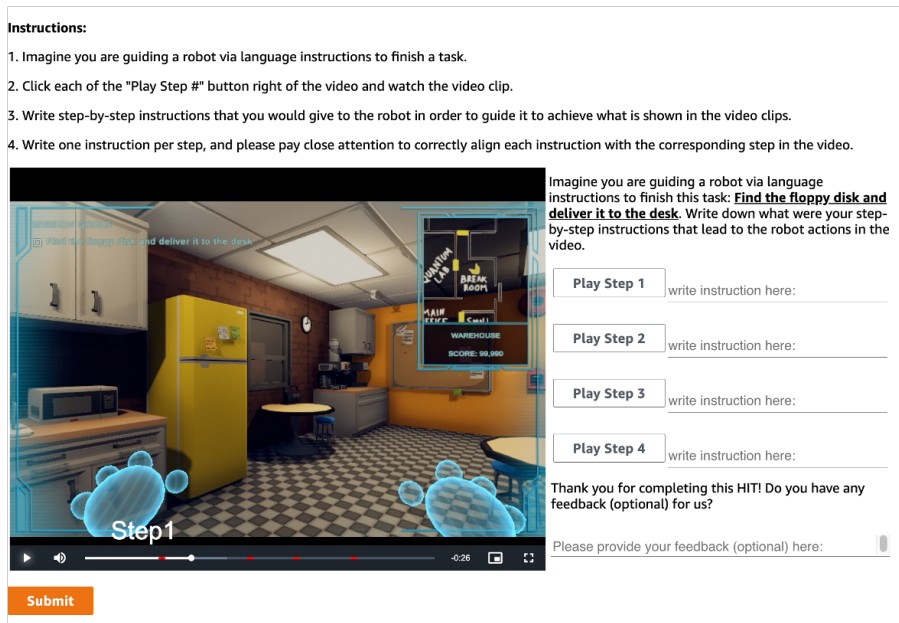

(a) User interface for collecting human instructions. Annotators watch the expert demonstrations and write down instructions to guide the robot.

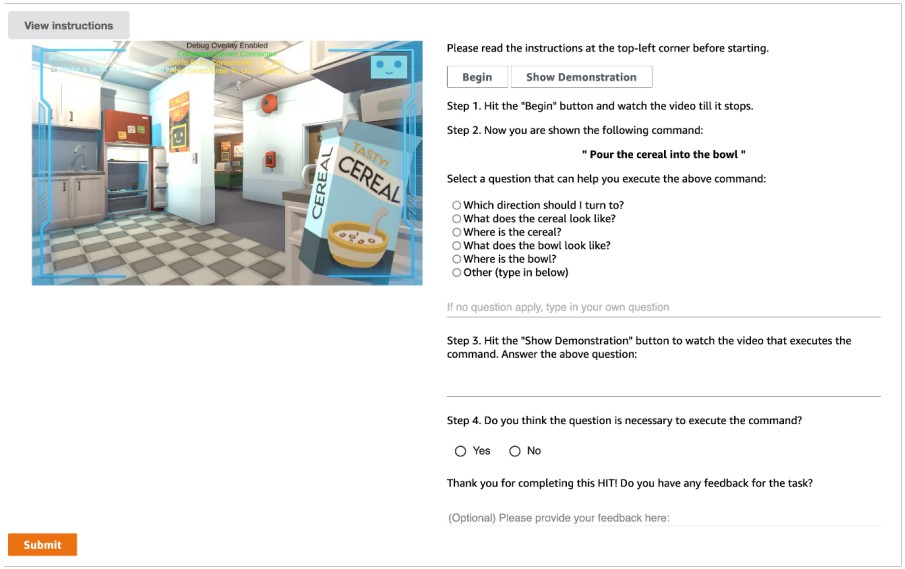

(b) User interface for collecting questions and answers. Annotators input their questions and answers to help the robot complete the mission.

Figure 9: User interface for data collection.

### A.3.2 Questions Generation

Similar to [11], the questions choices in the second stage of annotation are generated using predefined templates. Given an instruction, we extract nouns and insert them into the templates to construct questions. The nouns extracted correspond to query objects that the agent need more information to complete the mission.

In particular, we consider three types of questions, related to the location and appearance of the query object $o$, and the relative direction between the agent and the target position. The templates for each question are defined below:

1. Location: where is $o$?

|        | # scene | # task | # session | # instr | # dialog |
|--------|---------|--------|-----------|---------|----------|
| Train  | 6       | 2661   | 7983      | 40443   | 40105    |
| Valid  | 6       | 383    | 1149      | 6120    | 6128     |
| **Total** | 6    | 3044   | 9132      | 46563   | 46233    |

Table 4: `AID` Dataset Breakdown. # session represent the number of data sessions. Each data session corresponds to one language annotation on a given task. # instr stands for the total number of instructions in each data split. # dialog is for the total number of question and answers.

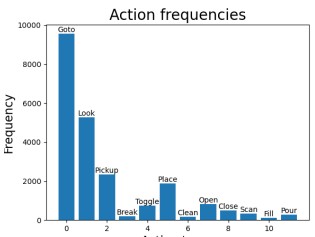

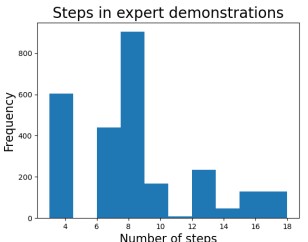

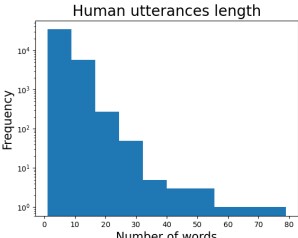

(a) Frequencies of action types.     (b) Steps in expert demonstrations.     (c) Length of human utterances.

Figure 10: Statistics of the `AID` dataset.

2. Appearance: what does $o$ look like?

3. Direction: which direction should I turn to?

4. Reference: which $o$ are you referring to?

### A.3.3 Oracle Answers

Human language has a lot of variety and complexity, and thus can be quite challenging for the agent to understand. Alternatively, we can provide templated language to the agent as a starting point for language understanding. To this end, in addition to asking human annotators to answer the questions, we also build an oracle that can answer the questions by extracting information from the simulator metadata: (i) To answer the location question, we compute the direction of the object relative to the agent and the viewpoint that is closest to the object in the room. To help detect small objects, if the object is held by a container, we also provide the information of the container and other objects that are also held by the same container as landmarks; (ii) To answer the appearance question, we directly extract object shape, color and material information from the simulator metadata; (iii) For direction question, we compare the agent's location at the end of the sub-task with its initial location to provide answers for the agent's moving direction. After we have the necessary information, we use language templates to generate the answers. Example templates include:

1. Location: The $o$ is to your [*direction*] in/on the [*container*] next to the [*landmark*] in the [*room*]. It is closest to [*viewpoint*].

2. Appearance: The $o$ is [*shape*] and of [*color*]. It is made of [*material*].

3. Direction: You should turn [*direction*] / You don't need to move.

### A.3.4 Dataset statistics

Table 4, Figure 10 and Table 5 show the statistics of the `AID` dataset.

### A.4 Vision Dataset

Visual perception is integral for EAI agents to navigate and interact within their environments. Visually intelligent robots use visual cues to set navigational targets and build visual representations based on perceived object state transitions. To facilitate visual AI research within and outside the context of EAI, we are releasing a vision dataset based on Arena. The dataset is composed of 600k training images and 60k validation images spanning 336 unique objects from more than 160 semantically grouped object groups. This can be used for large scale computer vision research on Arena, e.g. object detection, object state classification, etc. The overall structure and contents of the dataset are illustrated in Fig. 11. The vision dataset is collected by programmatically configuring

| Mission Type | Train | Valid |
|:---|:---:|:---:|
| breakObject | 196 | 30 |
| clean&deliver | 161 | 29 |
| color&deliver | 29 | 4 |
| fill&deliver | 108 | 16 |
| freeze&deliver | 76 | 8 |
| heat&deliver | 42 | 13 |
| insertInDevice | 381 | 59 |
| pickup&deliver | 669 | 95 |
| pourContainer | 265 | 39 |
| repair&deliver | 131 | 18 |
| scanObject | 324 | 37 |
| toggleDevice | 279 | 35 |

Table 5: Distributions of mission types across training and validation splits of the AID dataset.

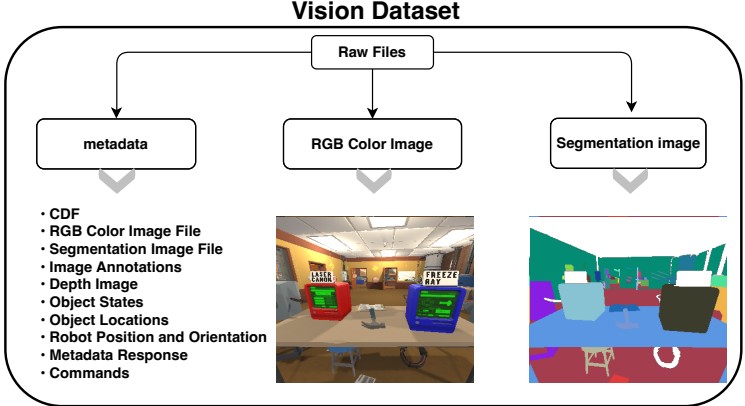

Figure 11: Overview of the vision dataset. The dataset consists of RGB and ground truth segmentation images in png format, along with a json file for each image with the metadata information.

CDFs and initializing scenes with all objects spawnable in the Arena environment. We then navigate to each of them to capture images from different perspectives and distances.

### A.4.1 Vision Data Generation

The image data collection strategy was simple and straightforward - the goal was to collect data spanning all possible objects spawnable in Arena from different distances, views and perspectives of the said object. This was done by following the below steps:

1. Initialize a game mission using the CDF file with required objects.

2. After initialization, we issue commands to navigate to all the different objects present in the scene using the primitive navigation actions and capture the first party view.

3. We then capture images by issuing interactive commands as per the affordances of that object at the time of capture, which are retrieved from the game metadata response.

4. We also capture additional perspectives by taking a random walk around the object using basic trigonometry.

We repeat the above steps with various missions and objects with lights on and off for different lighting conditions. All the commands, object states and their locations in the scene, along with a host of other useful information are logged in metadata files. We provide the RGB color images, segmentation maps and associated metadata for each image. The statistics of the dataset are illustrated in Fig. 12.

**Dataset Validation.** To validate the quality of the ground truth segmentation and bounding box annotations for modeling, we collect 1245 images containing instances spanning all object classes

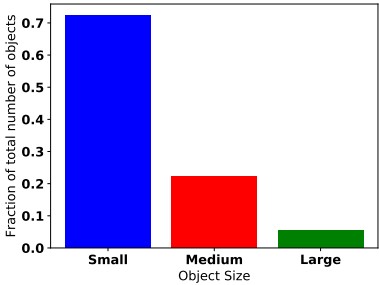

(a) Vision data object size distribution.

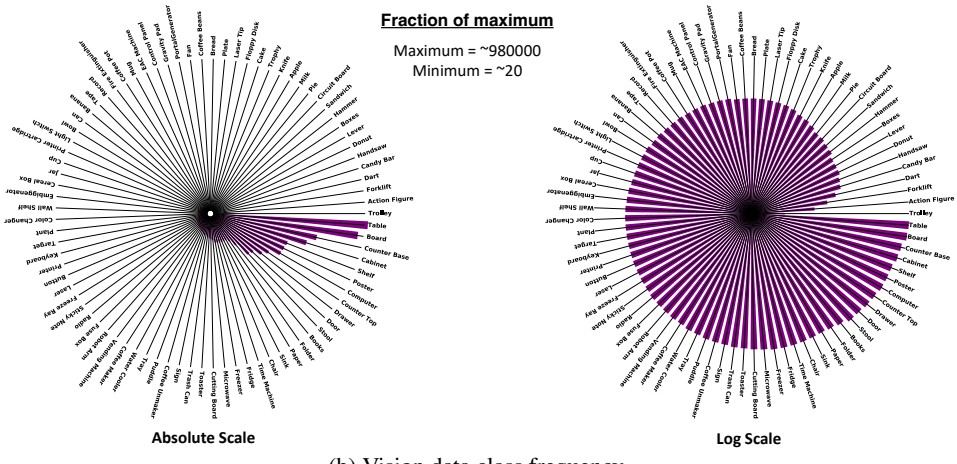

(b) Vision data class frequency.

Figure 12: Vision dataset statistics

and annotate them with bounding boxes and associated class names. Since Arena features fantastical objects (like *Gravity Flippers*), we also provide example images for each object class to familiarize the annotator with the appearance of all classes. The images are then validated by annotators according to the class names into one of the following 4 categories (with results provided in parenthesis):

1. Yes, the object is fully contained within the bounding box (*79.7%*)
2. Yes, the object is partially contained within the bounding box (*16.3%*)
3. No, the object is not within the bounding box, but is in the image (*1.6%*)
4. The object is not in the image (*2.4%*)

For category 2, the objects are present only partially in the image because they are only partially visible in the robot's egocentric view at the time of image collection. Overall results show that the ground truth segmentation and bounding box annotations have very little noise and are of high quality to be used for training and evaluation of visual perception models.

## A.5 Baseline Models Details

### A.5.1 Neural-symbolic Model

The architecture is shown in Figure 13. In particular, the multi-modal Transformer model uses the previous actions, previous visual observations and the full utterances of the episode to predict the next action type and object class before a *stop* token is predicted as the action. For the visual observations, we first extract features using the backbone of the vision model before sending them into the transformer. During training, after each look around action, the model uses the observations from different directions to predict rotations: the number of rotation depends on which image out of the four panoramic images is used by the following actions in the expert demonstrations. During

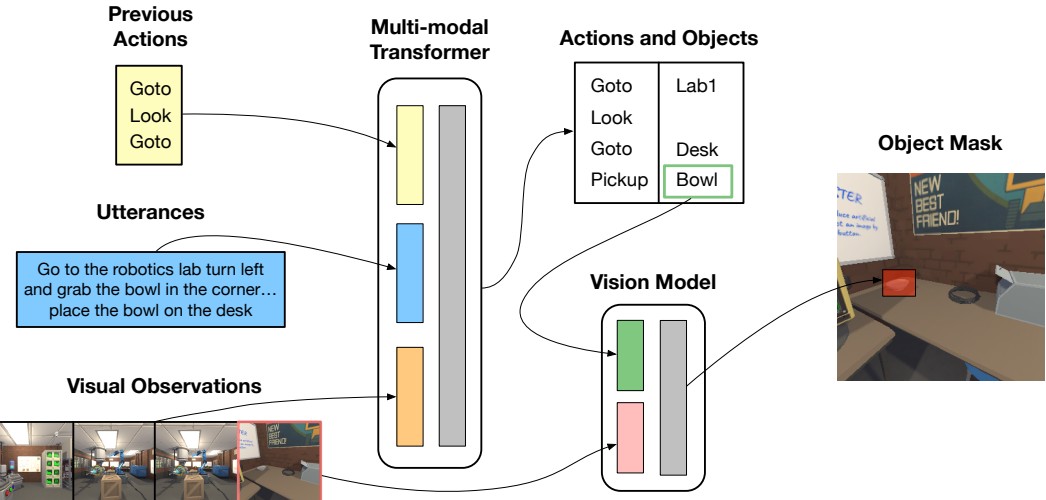

Figure 13: Model architecture for the Neural-Symbolic approach which uses unshared modules to predict actions and masks, and heuristic rules for visual grounding.

inference, the vision model detects all the objects from the current image, and uses the predicted object class to generate the mask for the corresponding object. Note that if the vision model detects multiple object instances of the same class, we select the one with the highest likelihood. We trained the transformer model for 20 epochs with a batch size of 2. The training takes around 10 hours on 1 NVIDIA A10 GPU. We use the Adam optimizer with a learning rate of $1e^{-4}$ for the first 10 epochs, and $1e^{-5}$ for the remaining epochs.

The Neural-Symbolic model is composed of a separate visual component that performs instance level segmentation of the first party view and/or surrounding images of the agent. We use the vision dataset to train a Mask-RCNN image segmentation model [14]. We process the object classes in the vision dataset to group them into 86 semantic classes including the background class. The model takes as input an RGB image and predicts masks for all object instances (across the 86 object classes) present in the image along with their class labels and confidence scores. The masks and object class predictions are then used in conjunction with the action and object predicted from the multi-modal transformer to give a single mask prediction for the interactable object. The vision model is trained for 22 epochs with a learning rate of 0.00125 and a weight decay of 0.001 with the SGD optimizer, decaying the learning rate after 15 epochs by a factor of 0.1. We use a global batch size of 16 and trained the model on 2 Tesla V100 GPUs, and the training takes around 24 hours. We evaluate the instance segmentation model using the standard COCO evaluation metric, the Mean Average Precision (mAP). The mAP metric is calculated by averaging the precision at Intersection over Union (IoU) thresholds ranging from 0.5 to 0.95 in steps of 0.05. The score of the predicted instance class is not taken into consideration, and instead a cap is applied on the number of maximum detections per image (chosen to be a 100). To be able to choose an operating point for our end-to-end model and also tune for recall, in addition to the standard COCO mAP metric, we also provide a score and iou thresholded precision metric, which we call *t-mAP*. This is calculated by thresholding the output instance class scores at [0.05, 0.1, 0.3, 0.5, 0.7], and thresholding the IoUs at [0.1, 0.3, 0.4, 0.5, 0.75, 0.8] for all 30 (score, IoU) combinations and then averaging them. We present both metrics for all objects, as well as metrics categorized for small, medium and large objects. The results are provided in Table 6.

### A.5.2 VL Model

For training, we use a learning rate of $1e^{-6}$ for the backbone and mask prediction head, and $1e^{-4}$ for the action prediction head (chosen empirically). We use the Adam optimizer with a weight decay of $1e^{-5}$ and a learning rate decay of $0.1$ after 35 epochs. The model is trained for 65 epochs with a global batch size of 64 (16 per GPU) on 4 Tesla V100 GPUs, and the training takes about 7 hours.

| Category | Area ($px^2$) | COCO mAP | t-mAP |
|----------|---------------|----------|-------|
| Small    | 0 - 1296      | 37.63    | 91.49 |
| Medium   | 1296 - 9216   | 60.41    | 92.15 |
| Large    | 9216 - 90000  | 64.72    | 84.91 |
| Overall  | 0 - 90000     | 46.03    | 89.5  |

Table 6: Image segmentation results for the vision model used in the neural-symbolic approach. We present the metrics for small, medium and large objects.

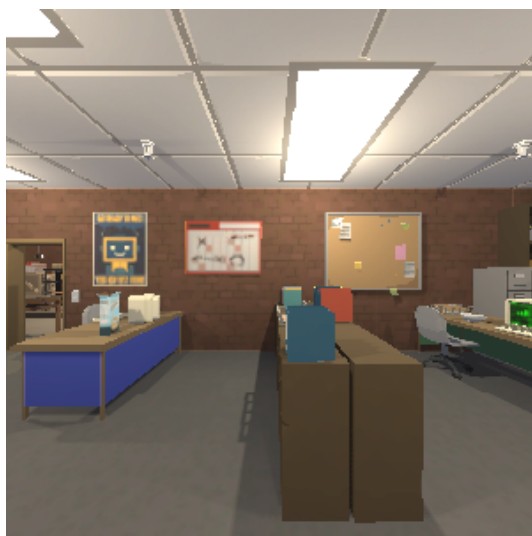

Figure 14: There are two tables in this image, a blue table on the left and a green one on the right, leading to challenging disambiguation tasks for the agent if not guided adequately.

### A.5.3 Encoding Questions and Answers

We use two tokens to efficiently encode the predefined questions, including (i) a token representing the question type (i.e. *loc* for questions asking about object location, *app* for object appearance, *dir* for direction and *ref* for reference), and (ii) a token for the target object in location, appearance, referential questions. For example, a question asking about the appearance of a microwave will be encoded as *app microwave*. Since the direction question is asking about the direction and not related to an object, the second token is not necessary and thus omitted for this type of question. For questions that are not of the predefined types, we include the whole question and answers.

### A.6 VL Model Error Analysis

While the VL model demonstrates improved performance using an end-to-end method that generates actions and masks simultaneously, we identify key insights, sources of errors and failure points of the end-to-end architecture by analyzing the robot trajectory for 25 missions:

1. Some missions fail because of failure to visually ground the natural language instruction to a single object instance for interaction in the case of multiple instance candidates. E.g. in Figure 14, there are two *desks* - blue on the left and green on the right. If the natural language instruction is not descriptive enough, the model fails to visually ground the desk of interest. On analysis, we notice that in the *<x>&deliver* mission type, the agent is often able to complete the *<x>* goal and fails at the *deliver* goal because of ambiguity in language (*"The desk is right in front of you with the monitor on it"*), or model visual perception failures to disambiguate between multiple instances of the same object.

2. Sometimes, if the agent veers off its path due to a wrong prediction (e.g. if it goes to a wrong room), the subsequent instructions to find an object in the room steers the agent down a path to infinitely rotate about its position to try finding the object ,i.e., the agent is trying to respond to the next instruction that assumes successful attainment of the correct previous state. This can

potentially be solved by better error correction mechanisms in the model to identify such loopy states and exit them to explore other options in the trajectory, or to execute a strategy to backtrack to previous instructions. Since the current baseline models use the readily available answers in the dataset, this can also be improved in models that are designed to ask questions based on its real-time current visual observations.

3. Since this is an end-to-end model, the language and visual features are jointly encoded and trained end-to-end from input language and images to output masks and actions. Some errors arise because the model is attending more to the visuals in front of it rather than the language input. E.g. if the robot is standing very close to a cabinet door, or a shelf with a coffee pot, even though the language instruction was to *"go to the laser cannon"*, it will try to *Open* and *Close* the cabinet door, or *Pickup* the coffee pot, respectively, which the model has learnt from the data.

4. Some more sources of error distributed in the model across all evaluation tasks include ambiguity in language, inability to model longe-horizon task instructions like *"turn right, go to the breakroom, heat the bowl and place it on the table"* - which consists of at least 9 steps, inability to detect masks for some very small objects, or for objects too up close, and failure to distinguish between very similar looking objects (like *Laser Cannon* and *Freeze Ray*). These are common issues between the Neural-Symbolic and the VL models.

5. We also did an ablation study by training only the mask prediction head and using the ground truth action predictions. Evaluating this model on the validation dataset yields a MSR of 71.89%. This simulates the scenario when the perfect actions are predicted to situate the robot in a position to predict a mask for an interactable object, thus evaluating only object detection and visual grounding of the natural language instruction. This result shows the model's difficulties in finding object locations in an end-to-end fashion, requiring the incorporation of better path planning modules that learn from multi-modal inputs.

