# OpenReview forum: "Alexa Arena: A User-Centric Interactive Platform for Embodied AI"
_NeurIPS.cc/2023/Track/Datasets_and_Benchmarks — NeurIPS 2023 Datasets and Benchmarks Poster_

### Official Review · Reviewer_Yxqr · 2023-07-20
**A platform where user can simulate commanding/guiding a robot by natural language to accomplish a task in indoor scenes**

**Rating:** 6
**Confidence:** 3
**Correctness:** Seems correct.

**Strengths:**

+ Interesting and significant idea. Indeed, natural language is reasonable for human to control/guide robotics to accomplish a task as a collaboration.

+ Both language input and key board/mouse action control.

+ Very rich accessories with carefully designed properties. Some are even unusual in daily life.

+ Wide-range task compositions and action space.



**Additional Feedback:**

Additional question: does the metric "Average number of robot actions (NRA)" include the conversation as one action? If so, how do you evaluate the efficiency of the dialogue? Is there any metric to measure how efficient the robot responds to a wrong/misleading/ambiguous human instruction/guidance? For example, when human deliberately tell the robot a wrong direction, can current metrics measure the efficiency of robot to find/correct this mistake?

**Clarity:**

Fairly written in my personal perspective. I think the writing flow is not clear enough and is a bit hard to read. For example, in Line 15, it writes "However, the current EAI platforms suffer from a set of limitations", without detailed explanation on what limitation.


**Documentation:**

Documents are fully included.

**Ethics:**

Can this simulator, in any way, be used/extended for military/killing-related tasks? Will such language guidance be expanded to killing command? Do the authors include any restriction to prevent such uses?

**Limitations:**

see above.

**Opportunities For Improvement:**

Since the core of this work is the platform, I personally think it is better to clearly present what the software can do to the audience in details.

- [authenticity] In the demo video, the robot's responses seem to be generated automatically. How does it happens? Don't the dialogue data, including both human's & robot's, need to be written by the user?

- In Appendix, the authors describe a lot of fancy scenes, e.g., Freeze the soda can with a freeze ray. Are these already included in your data? Why the demo does not include these?

- [Detailed functions] Can the authors provide some screenshot videos of the data making process using this platform with showing all interface (the keyboard/mouse/typing)? Meanwhile, it seems the platform has many kinds of settings on the dialogue (appendix), could you record these into the video? Also, if possible, would the author record a screenshot video when using this platform as evaluation, especially the "online" one (both success & failure) ?


**Relation To Prior Work:**

Yes. But the main comparison table is presented in the appendix.

**Summary And Contributions:**

This paper present a platform that can be used to simulate a robot accomplishing a compositional indoor task, such as "go to the kitchen and break a bowl to the ground" with a dialogue to have guidance from human. Such a simulator can be used to (1) generate data for related tasks, which can include (a) the RGB image in robotic's view (b) room map and robot location (is it correct?) (c) robot action (d) human-robotic dialogue, as well as (2) evaluate a robotic algorithm especially in an "online" way, that is, real human to chat with it and guide it. As claimed by the author, this platform allows both natural language input to construct the dialogue and key board/mouse to control the robot's actions, which are not included in existing works.

The authors also construct a dataset and set up a benchmark

Generally, the idea of this paper is interesting and could make sense for real scenes. The platform seem to be complete. But the reviewer has some concerns on (1) authenticity and (2) detailed functions after reading the paper and watching the demo video.

---

> ### Author Response · Authors · 2023-08-30
>
> Thank you for your constructive comments!
>
> > In the demo video, the robot's responses seem to be generated automatically. How does it happens?
>
> The demo video is an illustration of the capabilities of the platform for dialog-guided robotic task completion, not for demonstrating the human annotated data (where both human and robot's language are annotated by the user).
>
> > In Appendix, the authors describe a lot of fancy scenes, e.g., Freeze the soda can with a freeze ray. Are these already included in your data? Why the demo does not include these?
>
> The data includes tasks that includes atomic actions that utilize soda cans and freeze rays. The demo is an exemplar video to showcase the capabilities and the potential of the platform and is not exhaustive.
>
> > Can this simulator, in any way, be used/extended for military/killing-related tasks? Will such language guidance be expanded to killing command? Do the authors include any restriction to prevent such uses?
>
> We admit that via open-sourcing, this platform does not strictly prevent people from modifying it and even using it for malicious use cases. But we believe this has relatively low risks since the current resources we released do not contain elements that can be directly applied for military or killing-related tasks.
>
> > does the metric "Average number of robot actions (NRA)" include the conversation as one action?
>
> The NRA metric includes only the executed actions and does not include the conversation as an action. The metric is used to illustrate the increased efficiency with which the robot can execute a given task, when it is conferred with the ability to ask questions and clarifications with the human user. However, this is a good point that the quality of the dialog is an important metric to measure, which we can investigate in future work.

---

> > ### Comment · Reviewer_Yxqr · 2023-08-31
> >
> > Thanks for the authors' response. After reading the response, I maintain a positive attitude to this submission.

---

### Official Review · Reviewer_zBVW · 2023-07-21
**The proposed "Alexa Arena": A User-Centric Platform for Embodied AI Research Facilitating Gamified Robotic Tasks and High-Efficiency Data Collection" is interesting and valuable to the research community.**

**Rating:** 8
**Confidence:** 4

**Strengths:**

The "Alexa Arena” includes a valuable dataset and an interactive evaluation protocol, encouraging robust and authentic research. With the focus on assistive conversational agents, it has significant societal implications, bolstering the development of AI designed for human assistance. The provision of the platform and its rich dataset for public use highlights a commitment to open and collaborative research.

**Additional Feedback:**

In Table 2, the team “GauchoAI" seems to have the best MSR and user satisfaction, but why the team “SEAGULL” win the competition in the end?

**Clarity:**

* Yes, the submission is well written.

**Correctness:**

* Yes, the submission's claims are accurate, with a well-constructed dataset.

**Documentation:**

* Yes, the manuscript provides comprehensive details on the annotation process based on Amazon AMT, along with guidelines on ethical and responsible use. Furthermore, the dataset has been made openly accessible on GitHub and has been used in the competition.

**Limitations:**

* The authors have effectively taken into consideration the shortcomings and possible adverse consequences of their research.

**Opportunities For Improvement:**

* In Table 3, can you explain why the mission success rate for "color&deliver" is notably low (consistently below 3%)? Does this indicate an issue with how the task is defined, or is it largely due to its heavy reliance on extended contexts?

* Abbreviation should be explicitly explained before used, such as HRI in line 17;

* It would be beneficial to provide a comprehensive comparison of existing EAI benchmarks, specifically detailing the performance and capabilities between benchmarks such as META's Habitat and AI2-THOR

**Relation To Prior Work:**

* It will be nice to make a detail comparison table between the existing EAI benchmarks, such as benchmarks on META's Habitat and AI2-THOR.

**Summary And Contributions:**

The paper present "Alexa Arena", a user-centric platform developed for Embodied AI (EAI) research. The platform, characterized by multi-room layouts and many interactable objects, aims to facilitate the development of gamified robotic tasks that general users can easily engage with. This fosters high-efficiency data collection and EAI system evaluation.

Key contributions include:
1. The development of Alexa Arena, a platform designed specifically to aid in the construction of assistive conversational agents, by providing an immersive and interactive environment to study the interaction between humans and embodied agents.

2. The creation of a substantial dataset with over 9,000 dialogue sessions and 46,000 human-annotated instructions. Each session simulates the communication between a human user and an embodied agent engaged in a household activity.

3. The introduction of an interactive evaluation protocol, along with offline evaluation, allowing real-time progress observation and feedback provision. Baseline models for both evaluations are also presented.

---

> ### Author Response · Authors · 2023-08-30
>
> Thank you for your constructive comments!
>
> > In Table 3, can you explain why the mission success rate for "color&deliver" is notably low?
>
> Upon analysis, the "color&deliver" mission success rate is affected by the "deliver" part of the task, due to the high ambiguity in describing the accurate receptacle of delivery. The receptacle of delivery is often a table, which is not easily disambiguated by the model from human language, since there are many tables in the scenes which require careful disambiguation. An example is provided in the appendix A.6 and Figure 14.
>
> > It would be beneficial to provide a comprehensive comparison of existing EAI benchmarks
>
> We do have a table (table 3) in the supplementary material that compares Arena with other EAI platforms.
>
> > In Table 2, the team “GauchoAI" seems to have the best MSR and user satisfaction, but why the team “SEAGULL” win the competition in the end?
>
> The results shown in the table are from the semi-final stage of the Simbot Challenge, not the final event. After the semi-final stage, the teams were provided more time to improve their models, and the winner was selected based on their performance in the final event, where along with the MSR, human judges were also used to judge the quality of their interactions with the finalist models.

---

> > ### Comment · Reviewer_zBVW · 2023-08-31
> >
> > Thanks for the response. I keep my score.

---

### Official Review · Reviewer_PEw7 · 2023-07-21
**A user-friendly, conversational robot simulation platform along with a benchmark and a dataset**

**Rating:** 7
**Confidence:** 4
**Correctness:** The methodologies and evaluations see…
**Clarity:** The paper is polished, very well-writ…

**Strengths:**

1. The simulation environment seems user-friendly and engaging, which can help with better data collection.
2. This framework provides the ground work to test compositional learning capabilities of embodied AI models by considering states for object.
3. The problem this work tackles is important since it enables easier data collection in human-robot interaction scenarios where it is usually difficult and requires controlled environments to collect data.
4. The paper is presented very well and is easy to follow.



**Additional Feedback:**

The gamification of the platform can be a double-edged sword: it makes the data collection **easier** and more **fun**, but at the same time it makes the sim2real transfer more **difficult** (e.g. freeze ray is not practical in the real-world scenarios). What are your thoughts about this?

**Documentation:**

The dataset and the code are publicly available on GitHub. The authors provide thorough documentation.

**Ethics:**

The authors obtain an IRB for their work and include a brief discussion on ethical considerations.

**Limitations:**

The authors discuss the limitations of this work, including the difficulty of evaluating online models without a human-in-the-loop.

**Opportunities For Improvement:**

1. The Arena Interactive Dialogue (AID) Dataset contains only two splits: training and validation. This would limit the ability of this platform to evaluate models on held-out test sets. Would you please clarify why you split the dataset into two folds only?
2. Would you please explain why did you decide to use an LSTM in the end-to-end vision-language model instead of a transformer-based model?


**Relation To Prior Work:**

The authors do a very good job to relate their work to prior works in the areas of embodied AI, language-guided navigation and task completion, and task planning using large language models.

**Summary And Contributions:**

The authors present a user-centric, gamified, and conversational embodied AI platform with enhanced user experience to make HRI data collection easier. The authors also introduce a dataset of more than 9K dialog sessions and 46K human-annotated instructions. Moreover, they provide a web-based interface for evaluation where users can provide instructions and monitor the progress.

The gamification is done by presenting tasks as games and employing scoring systems and achievements. To combat the problem of overfitting and having generalizable models, the authors design tasks that can be completed in different ways with compositional and causally interconnected state changes.
To enhance the user experience of this platform, this platform uses engage-enhancers such as a scoring system, animated actions in a continuous way, and a unique user interface.

This platform contains 336 unique objects each with a set of properties, called affordances. There are a total of 14 properties. Two types of actions are possible in this platform: user interactions (e.g., communicating with the user via dialog) and robot physical actions (which include navigation and 11 object interaction).
To evaluate dialog-guided agents, this paper presents the dialog-guided task completion benchmark for indoor object interaction tasks. Each task has initial states, final goal states, and human instructions in the form of natural language.

Moreover, this work introduces Arena Interactive Dialogue (AID) Dataset collected via crowd-sourcing. This dataset contains ground-truth action trajectories, human natural language instruction, and corresponding questions and answers in the form of dialog. There are a total of 12 types of tasks in this dataset. The game tasks are generated by sampling from initial states and task goals, and then a planner generates the expert demonstrations. The authors use Amazon Mechanical Turk to collect language dialogue in two stages. First, the participants provide instructions on how to finish a task. Next, the participants provide questions (and their answers) to better perform the task.

The authors implement two baseline approaches to perform tasks in this platform: a neural-symbolic approach using Episodic Transformer (ET) and an end-to-end vision-language model. In the second approach, the language and vision embeddings are passed to a joint vision-language encoder that goes to two heads then: the task planning head using an LSTM and the mask prediction head to predict the mask referring to the object.
To evaluate the models, the authors use two metrics, including mission success rate (MSR) and the average number of robot actions (NRA) to measure efficiency.

---

> ### Author Response · Authors · 2023-08-30
>
> Thank you for your insightful comments!
>
> > Would you please clarify why you split the dataset into two folds only?
>
> The training and validation splits are provided to aid model development. Meanwhile, we have also created a test set and it is hosted as a public challenge on eval.ai https://eval.ai/web/challenges/challenge-page/1903/overview
>
> > Would you please explain why did you decide to use an LSTM in the end-to-end vision-language model instead of a transformer-based model?
>
> The provided model is an exemplar baseline model used to benchmark the dataset and dialog-guided task completion.  From our experiments with using an LSTM model to predict the action-object pair given user instructions, we observed that we could achieve a good (~70%) accuracy, which led us to adopt the same for the vision language model. However, we also believe the transformer architecture could have a competitive advantage under the same framework which we have introduced. We leave this as an exercise to readers or adopters of the platform to experiment with other architectures that improve vision-language understanding of the models.
>
> > The gamification of the platform can be a double-edged sword
>
> This is a very valid point. Throughout the design of the platform and the game missions, we aim to strike a balance between real-world transferability, progress towards procedural, interpretable solutions and increased research-phase user engagement, which will aid in long term gains in the field of EAI. To ease the data collection effort, we introduce gamification to EAI, which is achieved by presenting tasks or user interactions in the form of games, by introducing scoring mechanisms, achievements and streaks to stimulate and engage users, thus promoting and more importantly, retaining their participation. While some of the missions themselves are not transferrable to the real-world, the focus should be on the algorithmic improvements and gains which can be brought about by using the EAI platform, which can be transferrable to real-world scenarios with similar complexity (e.g. replace freeze ray with a microwave or a stove for heating elements in home automation), which we believe is the true potential of this work.

---

### Official Review · Reviewer_A3RV · 2023-07-24
**Alexa Arena**

**Rating:** 7
**Confidence:** 5
**Clarity:** Yes, besides there being some missing…

**Strengths:**

Significance: This is a new platform for studying language use in interactive, embodied scenarios. In comparison to existing platforms (e.g., ALFRED, CerealBar), it adds a few additional features, such as gamification to make the task more interesting for human annotators, a realistic environment, more hierarchical "missions" to complete, and more complex world dynamics (e.g., multiple ways to accomplish a single goal).

Relevance: This is very relevant to much of the current work in embodied language use. The later experiments on continual learning through interaction a relevant to a few recent works on learning from human feedback in embodied, interactive scenarios.

Quality of research: The game is well designed, with additional features on top of what existing platforms already provide. Some of the interaction affordances (e.g. highlighting an object) are also novel for these platforms. The paper also includes several experiments; I am particularly interested in the experiments on continual learning through interaction.

Ethical and social implications: This is not discussed much in the paper, beyond the common motivation that agents which can collaborate with people in embodied scenarios have many applications, e.g. in accessibility.

**Additional Feedback:**

N/A

**Correctness:**

The collection of the static dataset is somewhat confusing and unnatural. However, as evaluation is focused on online human-agent interaction, it could work okay for training and development.

More details on the setup for the online evaluation would be useful, particularly in terms of the population of users, how users might change their behavior over time, etc. (see questions above).

**Documentation:**

Because the online human-agent evaluation is "gold standard" for this type of task, I am not sure whether there is a plan for this to become broadly available to the community, or if it is reserved for those submitting to a shared task. Is code available that could be used to replicate the setup for online evaluation, if the authors do not plan to continuously/publicly host this evaluation?

**Limitations:**

Yes

**Opportunities For Improvement:**

There are some details missing from the paper that I was confused about:
* Are user interventions (e.g., taking over the system's tasks) actually possible in Alexa Arena? Or, is the user just involved via the dialogue interface? What kind of feedback can users provide (which those participating in the shared task can use for continual learning)?
* How are missions chosen and designed?
* How are the videos segmented for the static data annotation?
* What is a "failed step" (L225)?

Some details about the online evaluation are missing, which I believe are quite important to consider:
* How does this evaluation take into account variation between people and adaptation that might happen within an interaction?
* How are users changing their behavior week after week? Does this affect the impressions the users have of the models, or are the models genuinely improving? (See Kojima et al. 2021 and Suhr and Artzi 2022 for discussion on the confounder of user behavior change)
* Showing the distribution of Likert scores in Table 2 would be more informative than an average.
* Are the 9 missions in the online evaluation training portion fixed? Are the environments themselves fixed? If so, what is ensuring that the model doesn't just overfit to these action sequences during the continual learning phase?
* How do the proposed methods perform continual learning?
* There are details missing about the setup for the online evaluation. Who is involved -- the same workers as during the static data collection? Is this designed to ensure consistency across weeks and between teams?

Some of the aspects of the task and data are a bit unusual:
* The way dialogues are annotated is relatively unnatural. As the data is mostly only proposed for learning, and evaluation focuses on online human-agent interaction, this not necessarily a problem, but it still is a bit curious that dialogues are invented after actions are generated.
* Evaluation using the static dataset, while not the focus of the work, is a bit strict of an evaluation metric, as it presumably does not allow for error recovery, and any deviation from the gold action sequence makes the text irrelevant to the system's behavior.

**Relation To Prior Work:**

Yes

**Summary And Contributions:**

This paper proposes a new platform for studying embodied language understanding via a collaborative game that requires following sequences of instructions. The paper also includes a dataset of static dialogues constructed on top of the platform, and reports results on an experiment with online human-agent dialogues where participants in a shared task deploy and update models across multiple rounds of user interaction.

---

> ### Author Response · Authors · 2023-08-30
>
> Thank you for your insightful comments!
>
> > Are user interventions (e.g., taking over the system's tasks) actually possible in Alexa Arena?
>
> Currently the user can only provide natural language intervention via the dialog interface. But given the APIs for robot atomic actions, it is also easy to enable user interventions on robot actions. For example, the "highlight" command can be used by the agent to ask for visual confirmation from the user, which the user can then clarify using natural language.
>
> > How are missions chosen and designed?
>
> The missions were designed to cover good variety of robot actions, object types, object state changes, and the tasks also need to be intuitive and meaningful for human users.
>
> > How are the videos segmented for the static data annotation?
>
> The videos are segmented via grouping adjacent actions into steps. Specifically, we have defined a set of templates for high-level actions, and the grouping are done via randomly applying the templates on the whole action sequence. For example, one high-level action "heat with device" has a template action sequence of "open(device), place_in(device), close(device), turn_on(device), turn_off(device), open(device)".
>
> > What is a "failed step" (L225)?
>
> A failed step is defined as a step that has not been successfully completed in the simulator. This can be obtained from the metadata after every action execution. For example, toggling a device that cannot be toggled is a failed step.
>
> > How does this evaluation take into account variation between people and adaptation that might happen within an interaction? ...
>
> We refer readers to another paper which better details the Simbot Challenge : Shi, et al. "Alexa, play with robot: Introducing the first Alexa Prize SimBot Challenge on embodied AI." arXiv preprint arXiv:2308.05221 (2023).
>
> > Are the 9 missions in the online evaluation training portion fixed? Are the environments themselves fixed?
>
> While the 9 missions were fixed during the "seen" phase of the challenge (i.e. the teams knew the contents of the games when they were allowed to make developmental changes), there was also an unseen phase of the challenge, during which the teams were not allowed to make changes to their code, and their models needed to solve unseen missions in unseen environments. The 9 missions and the associated results are provided as an illustration of the methods in which the platform could be used for iterative development of EAI.
>
> > How do the proposed methods perform continual learning?
>
> Continual learning is demonstrated by the way in which the models incorporate user feedback into the training process. However, the platform can also be used for active learning in which models are updated in real-time using real-time user feedback and online evaluation.
>
> > There are details missing about the setup for the online evaluation. Who is involved — the same workers as during the static data collection? Is this designed to ensure consistency across weeks and between teams?
>
> The workers for data collection are different from the users that test the systems during online evaluation. We ensure that the usage is consistent across teams by ensuring that a user is randomly routed to a team's system. This ensures that all teams get roughly equal parts of the total number of users in the given time frame. All users are also provided with the option to engage in a tutorial to get familiar with the game prior to starting the evaluation.
>
> > The way dialogues are annotated is relatively unnatural.
>
> The current annotation set up is a trade-off between cost and data quality. This method enables the initial bootstrap for data collection, while also having the annotators adhere to the robot capabilities in the platform. More open ended forms of dialog annotation would be explored in future work.
>
> > Evaluation using the static dataset, while not the focus of the work, is a bit strict of an evaluation metric, as it presumably does not allow for error recovery, and any deviation from the gold action sequence makes the text irrelevant to the system's behavior.
>
> Yes, the metric is strict and does not include an interactive evaluation. As mentioned as a part of future work, to overcome this limitation, we aim to incorporate a model-based user-simulator, to which an oracle provides relevant answers during task execution in the offline phase.
>
> > Because the online human-agent evaluation is "gold standard" for this type of task, I am not sure whether there is a plan for this to become broadly available to the community...
>
> Since the first Simbot Challenge has ended now, we are not currently hosting an interactive human-agent evaluation platform. However, a similar evaluation workflow could be set up with the web-based user interface that we have open-sourced with the code (Figure 2).

---

### Official Review · Reviewer_WUL3 · 2023-07-28

**Rating:** 1
**Confidence:** 5
**Correctness:** No. This manuscript is misleading abo…
**Clarity:** No. The actual platform is not being …

**Strengths:**

Current platforms are very limited in scope.

**Additional Feedback:**

The Arena platform, the main contribution, is not made publicly accessible. The publication never mentions that the code on GitHub only consists of wrappers around a closed source Arena executable. One with this extreme license the nature of which is also described in a misleading way both in the publication and on GitHub. A license to which no university or corporate legal department would agree to.

This is the main contribution, in terms of what the title, abstract, and body of the manuscript claim.

Why not release it? It's hard to see what value advertising something that no one else can use provides.

**Documentation:**

N/A

**Ethics:**

No.

**Limitations:**

My biggest concern is that this work is impossible to use because the main contribution, the platform, is closed-source under an extremely restrictive license that gives away all IP to Amazon and prevents any future research and any scrutiny of the current work.

While the abstract states "We make Alexa Arena publicly available to facilitate research in building assistive conversational embodied agents." This is false. The platform is not public. The code that wraps the closed-source by-request-only executable for the platform is made available.

Moreover, this fact is hidden and not revealed in the publication. Nowhere in the publication is it made clear that a closed-source executable is needed to run this work. That the executable is only available by request. And that it involves an extreme, never before used, and unprecedented license. It is stated in one field in OpenReview. But, the implications of the "executable" being closed are not obvious: one would expect this to be a minor component, with the rest being open as advertised, not the entire platform itself.

A license which virtually all researchers, academic or industrial, would be forbidden by their legal office from using. Practically, it means that while this work is interesting, it is of extremely limited utility. No one can build on it but the original authors and everyone who works on it gives their IP to amazon.

1. The license gives away all IP to Amazon

While the data of the dataset is under a standard license, CC-BY-NC 4.0, the code of the dataset (equally essential) appears to be under a very non-standard license. One which has never appeared before in any project or dataset; indeed, this is the only time it has appeared online at all.

To use the platform one must get an "Arena executable". This is not open source, it is governed by the https://raw.githubusercontent.com/amazon-science/alexa-arena/main/ARENA_EXECUTABLE_LICENSE and it can only be obtained by personal request.

Section 5 states:

> 5. Reservation of Rights; Feedback

> The Materials are the intellectual property of Amazon or its licensors. Except for the rights explicitly granted to you in this License Agreement, all right, title, and interest in and to the Materials are reserved and retained by us and our licensors. If you make suggestions, ideas, or other feedback available to us relating to the Materials, we will be free to exercise all rights in such feedback without restriction and without compensating you.

A publication is by definition a suggestion, idea, or feedback related to the dataset. This means that anything anyone publishes includes a license to Amazon. Using this dataset would violate existing IP agreements that most researchers have. And most researchers would not agree to do research on this dataset given that it means Amazon has their IP. Moreover, most companies and universities would not allow their researchers to do this.

Worse. This isn't made clear anywhere in the publication or on the website. Indeed, GitHub states that the work is provided under an LGPL-2.1 license. This is highly misleading. The only note is that a separate non-commercial license exists, but this statement falls well short of revealing what the restrictions of this license are.

2. The license also prevents any future work

> 3. Limitations

> You may use the Materials only as expressly authorized under this License Agreement. You may not: (a) distribute, sub-license, sell, offer for sale, lease, transfer, publicly display, publicly perform, or otherwise provide access to any portion of the Materials to any third party; (b) modify, adapt, or create derivative works of the Materials; (c) circumvent or disable any copy protection, security, or other controls in the Materials; (d) reverse engineer, disassemble, or decompile the Materials (except to the extent applicable law doesn’t allow this restriction); (e) use the Materials with any software or other materials that are subject to licenses or restrictions (e.g., open source software licenses) that, when combined with the Materials, would require you or us to disclose, license, distribute, or otherwise make all or any part of such Materials available to anyone; (f) remove, modify, or obscure any copyright, patent, trademark, or other proprietary or attribution notices on or in any Materials; or (g) direct, encourage, or assist any other party to take any action prohibited by this License Agreement.

This means that no researcher can modify this platform. How can one add assets?  Modify robot actions? What if there is a bug? How can we change the physic model? The answer is that one cannot. The key contribution of this work, the platform, is in truth not released and not accessible.

I was dismayed to have to discover this and not have it clearly stated in the manuscript.

3. The main contribution is not available

It is hard to see the scientific utility of this platform if one cannot actually use it. Unless it is made available to researchers, both practically (not just as an executable) and under a license that researchers can use, the manuscript essentially describes what research only the authors can perform and no one else.

4. Limited and confusing object properties

Why are object properties so restricted? Virtually anything can break. Almost anything can be moved. Why for example art dart boards are not movable?

It looks like some objects are arranged in a hierarchy? "Computer_Monitor_01", "Computer_Monitor_Broken", "Computer_Monitor_New" are these all instances of computer monitor? Why is "Computer_Monitor_01" a receptacle and infectable while the others are not?

Similarly with paper trays, there are empty and full ones. How does the transition between them work?

Other object properties, while, in a game-world sense, are not logical to a human. A dart board is not a receptacle. Dart boards do not contain darts.

In general, the object and object interaction model appears to be very arbitrary and poorly specified. This very much limits how generic the dataset is and its long term applications to very narrow settings.

5. Reasoning

Very little reasoning is required. Almost all of the tasks are of the form "get object, perform action with object". This makes the questions very plain and simple to parse into a formalism like PDDL (which is precisely what the authors do). But it's hard not to see this as a missed opportunity. In a rich game-like environment why restrict yourself to such simple questions? While the dataset goes beyond previous work in raw scale, it does not innovate in terms of the complexity of the language used.

Instead of restricting this platform to a dozen simple mission types, why not define bigger missions like you see in a computer game (the need to avert some catastrophe or get through a dangerous lab), and have humans describe in natural language how a robot could solve those missions? Humans could describe the actions of the robot at different levels from the highest like "get through the lab without being electrocuted", to "that means avoiding using the broken coffee makers", to "touch the 2nd panel on the right as you enter the lab" assuming that one is not broken. This would lead to a far richer dataset without these predefined categories. One could of course imagine many similar setups.


**Opportunities For Improvement:**

See main review.

**Relation To Prior Work:**

Yes

**Summary And Contributions:**

A new navigation/manipulation platform to create language-driven robots in a game-like environment.

---

> ### Author Response · Authors · 2023-08-30
>
> Thank you for your insightful comments!
>
> First we apology for the confusing license terms associated with the released resources. The strict license terms were set before the Simbot Challenge. Now that the challenge is completed, we plan to make the resources more available to the community. Specifically:
> 1. Arena executable: we will provide downloadable link without request and approval, Apache 2.0 / CC-BY-NC 4.0 license
> 2. Source code for the Arena simulator: open-source under Apache 2.0 / CC-BY-NC 4.0 license
> 3. Arena wrapper code: open-source under Apache 2.0 / CC-BY-NC 4.0 license
> 4. Arena dataset (training and validation set): release under Apache 2.0 / CC-BY-NC 4.0 license
>
> We are currently going through legal validation process to release the above mentioned resources.
>
> > Other object properties, while, in a game-world sense, are not logical to a human. A dart board is not a receptacle. Dart boards do not contain darts.
>
> A receptacle is anything that can accept another object in one of its receptacle points. In this sense, a dart board is a receptacle for the darts. While these are non-human like in the real world, we believe this simplifies EAI development, as is the case in other EAI frameworks like Habitat and AI2-THOR, which we believe are simply programmatic simplifications which are present in simulation environment backends that do not affect user-experience in a dialog-guided setting. With regards to the hierarchical objects - as for the agent, we envision the agent learning nuanced affordances through human feedback or from interaction with the environment. We hope Arena is a test-bed for such nuanced EAI interactions. As for the human user, this will not impact user experience in a dialog guided interactive setting.
>
> > Instead of restricting this platform to a dozen simple mission types, why not define bigger missions like you see in a computer game ...
>
> This is a great point and we would like to extend the challenge and the dataset in future iterations to include more compositional and long range tasks. However, users can use the platform to create their own compositional datasets to evaluate models. As is evident from the baseline models provided in the paper, performance on the current missions types are also lagging in success rate. We believe that Arena is a suitable test-bed for developing and evaluating dialog-guided task completion across both simple and more complex mission types that you have described.

---

> > ### Comment · Reviewer_WUL3 · 2023-08-30
> >
> > ## License
> >
> > I appreciate that the authors are doing their best to secure a permissive license. And I am sympathetic that this is not an easy journey. But, at the moment, no such approval exists. The appendix continues to list the custom restrictive binary license. GitHub does so as well.
> >
> > The utility of this work depends entirely on releasing all of these resources under a permissive license. There are many scenarios under which such licensing would fail legal review. Or legal may demand a stricter license. Or a custom license as existed in the past.
> >
> > It seems entirely premature to accept this work at present, when there is a real risk that it will be highly restrictive and exclusionary. This is also an ethical concern: a requirement where one must request the dataset or binary, for example, strongly favors those who speak English and insiders who can make priority requests.
> >
> > Once accepted, there is no mechanism to ensure that the permissive licenses listed here will actually be the licenses under which the dataset is released. This is the only opportunity to ensure that the community actually has access to the work.
> >
> > I also want to note that because of the license, none of the reviewers could try the environment themselves. In my opinion this is critical for a positive score.
> >
> > That being said, the rebuttal did not address my substantive concerns about the research.
> >
> > ## Substantive review
> >
> > > Limited and confusing object properties
> >
> > The rebuttal engages with this question only minimally.
> >
> > It mentions one example I gave, that dartboards could be conceived as receptacles for darts. As a native English speaker, I disagree. But, I mentioned other examples, such as some computer monitors being receptacles while others not being receptacles. These arbitrary distinctions are a problem.
> >
> > Let me add more:
> > - Desks are not receptacles. A cutting board is not a receptacle. A laser is not a receptacle. Why can't you eat an apple slice even though you can eat an apple?
> > - Cake is not eatable, not even cake slices. Fruit pie is not eatable, but slices of fruit pie are.
> > - Soda cans are heatable, unless they are crushed, in which case they are not.
> > - Coffee beans are fillable? (I don't understand this at all) But cereal boxes are not?
> >
> > I could go on with over 100 such examples. The entire object property setup being used makes little sense. And this is a major problem. As models learn more about the world, and as language models are incorporated into robotics applications, the confusing and arbitrary properties used here risks producing incoherent results. The more I know about real objects, the worse I will perform on this benchmark, because none of the relationships I expect hold.
> >
> > > Reasoning
> >
> > Every task is of the form "get object, perform action with object". The rebuttal notes that others could come up with new tasks in this environment. This is fair, but does not address the point that this makes the benchmark very limited. We have many environments for fetching objects. And little for anything else.
> >
> > ## Overall
> >
> > Even if the license issues were resolved my score would not change. I have never been allowed to run this environment to see it in action and judge it. I believe no reviewer has. And the confusing object property assignments are a long term problem that should be addressed before release.

---

> > > ### Author Response · Authors · 2023-08-31
> > >
> > > > The utility of this work depends entirely on releasing all of these resources under a permissive license. There are many scenarios under which such licensing would fail legal review. Or legal may demand a stricter license. Or a custom license as existed in the past.
> > >
> > > We totally understand the reviewer’s concern related to the license issue. While we are actively working with legal on the license, we want to point out that the value of our work stands even with the current license. As shown in Appendix A.1.2, without making any modification to the underlying simulator, researchers can use the challenge definition format (CDF) files to configure game missions by specifying game scenes, robot location & states, object types, object location & states, and mission goal states. As an example, we were able to specify the game missions used in the Alexa Prize Simbot Challenge this way without modifying the platform.
> > >
> > >
> > > > Once accepted, there is no mechanism to ensure that the permissive licenses listed here will actually be the licenses under which the dataset is released. This is the only opportunity to ensure that the community actually has access to the work.
> > >
> > > We fully understand the reviewer’s urge to immediately make Arena more accessible to the community. Looking at the guideline of the Neurips conference, however, we believe we have the opportunity to make necessary changes “up to a year after the submission deadline” (https://neurips.cc/Conferences/2023/CallForDatasetsBenchmarks). We will adhere to this guideline and make the changes before the deadline.
> > >
> > >
> > > > I also want to note that because of the license, none of the reviewers could try the environment themselves. In my opinion this is critical for a positive score.
> > >
> > > We have included the script (fetch_arena.sh) to download Arena executable in the supplementary material and the reviewers are welcome to try it.

---

> > > > ### Author Response · Authors · 2023-08-31
> > > > **Response to Substantive Review**
> > > >
> > > > > Limited and confusing object properties
> > > >
> > > > The term receptacle is used to refer to anything that can afford to hold another object in one of its receptacle points. The point stated about darts should generally be applicable for all other objects, including desks.
> > > >
> > > > Regarding cakes, soda cans, coffee beans and other examples, we do believe that the backend definitions for the simulator do not affect user experience. In fact, this is a gamified setting that was designed so that the user and/or the agent can figure out how to complete the task given the environment they are in and the affordances of the objects around them.
> > > >
> > > > Regarding the concern that as models learn more about the real world and language models are incorporated into robotics, this doesn’t address the fact that generalized models are not applicable in every setting. We hope that the nuances in the object property set up would help EAI agent training and adaptation to different environments.
> > > >
> > > > >Reasoning
> > > >
> > > > The tasks also include scenarios of heating objects, coloring objects etc. which require reasoning about which set of actions to perform and which assets to use. While we did mention that we leave more complex and long range task datasets for future work, we would like to point out that even in this benchmark, the EAI agents are lagging in performance (MSR).
> > > >
> > > > We sincerely hope this clarifies your concerns.

---

### Author Response · Authors · 2023-08-30

Dear AC and all reviewers:

We sincerely appreciate the time and effort you’ve put in reviewing our paper. We are delighted to find that the reviewers have generally acknowledged our contributions, and we are thankful to all the insightful and constructive comments and suggestions. Please find our responses in each thread.

Best,\
Authors

---

### Decision · Program_Chairs · 2023-09-22

**Decision:**

Accept (Poster)

**Comment:**

This is a controversial paper, and reviewers (and ethics reviewers) do not agree on appropriate disposition.

- Pros: several of the reviewers have a positive impression of the paper, with the general sense that the presented testbed (the "Alexa arena") for exploring embodied AI in simulation has the potential to be a useful benchmark. The gamification of the human-robot interaction has the potential to improve uptake or at least participant experience, and the experiments presented make a baseline for future learning approaches. Overall, the game is relevant to current work on embodied language, and has some advantages over existing testbeds. If the license and ethical concerns weren't there (see below), this paper would be a suitable fit for the track.

- Cons: In its current form, the actual arena is essentially unavailable. What the developers have released is a wrapper around code which is accessible only by request from the developers, and use of that code is controlled by a rather draconian license. The terms of this license make reviewers unwilling/unable to experiment further with the testbed. Furthermore, the actual paper does not acknowledge this limitation, and instead describes the license which the author are currently seeking approval for. The authors point out, correctly, that in "select cases, requiring solid motivation," it is acceptable to release materials up to a year after acceptance; however, in this case the paper should acknowledge the current state.

- Ethical concerns included lack of acknowledgement of the potential problems with being English-centric; although the recommendation is that this be further discussed in the ethical considerations section, the changes consist of a single sentence stating that the authors have not looked at these issues. They also included concerns about the potential misuse of the arena to support unethical behavior on the part of the agents. One possible recommendation for this is, unfortunately, a _more_ restrictive license than that currently being sought. The suggestion that 'guard rails' be added to minimize the AI agent's ability to unknowingly violate the boundaries of ethics is a significant proposed change; the authors' response is that this is less necessary for their testbed, which does not contain items or scenarios of concern (e.g., bombs). This response doesn't address the ethical quandaries inherent to agents' language use (see, for example, Tay).

This is overall a good paper with significant flaws, and the question is whether those flaws represent dealbreakers for acceptance. The more substantial ethical recommendations are not within scope for a camera-ready revision, but represent possible failures that are rather more hypothetical; the nearer-term ethical concerns can be discussed by putting more attention into the Ethical Considerations section. The fact that the paper does not accurately represent the current state of licensing is a major flaw, but one that can be addressed for a camera-ready. I strongly encourage the authors to take the time necessary to address these points in any final or subsequent version of this manuscript, as well as other points and clarifications raised by reviewers.